



# Quantification and evaluation of atmospheric ammonia emissions with different methods: A case study for the Yangtze River Delta region, China

Yu Zhao[1,2*], Mengchen Yuan[1], Xin Huang[3], Feng Chen[4], Jie Zhang[4]

1. State Key Laboratory of Pollution Control & Resource Reuse and School of the Environment, Nanjing University, 163 Xianlin Ave., Nanjing, Jiangsu 210023, China

2. Jiangsu Collaborative Innovation Center of Atmospheric Environment and Equipment Technology (CICAEET), Nanjing University of Information Science & Technology, Jiangsu 210044, China

3. School of Atmospheric Science, Nanjing University, 163 Xianlin Ave., Nanjing, Jiangsu 210023, China

4. Jiangsu Provincial Academy of Environmental Science, 176 North Jiangdong Rd., Nanjing, Jiangsu 210036, China

*Corresponding author: Yu Zhao

Phone: 86-25-89680650; email: yuzhao@nju.edu.cn



**Abstract**

22        To explore the effects of data and method on emission estimation, two
inventories of $NH_3$ emissions of the Yangtze River Delta (YRD) region in eastern
China were developed for 2014 based on the constant emission factors (E1) and those
characterizing the agricultural processes (E2), respectively. The latter integrated the
detailed information of soil, meteorology and agricultural processes, and derived the
monthly information of emission factors and activity data. The total emissions were
calculated at 1765 and 1067 Gg, respectively, and agricultural activities (livestock
farming and fertilizer use) were estimated to contribute 74-84% to total emissions in
the two inventories. Clear differences existed in seasonal and spatial distributions of
$NH_3$ emissions. Elevated emissions were found in March and September in E2,
attributed largely to the increased top dressing fertilization and to the enhanced $NH_3$
volatilization under high temperature, respectively. Relatively large discrepancy
between the methods existed in northern Yangtze River Delta areas with abundant
croplands. The two inventories were evaluated through air quality modeling and
available ground and satellite observation. With the estimated emissions 38% smaller
in E2, the average of simulated $NH_3$ concentrations using E2 was 27% smaller than
that using E1 at two ground observation sites in the YRD region. At the suburban
SHPD site, the simulated $NH_3$ concentrations with E1 were generally larger than
observation, and the modeling performance was largely improved indicated by the
smaller NMEs when E2 was applied. In contrast, very limited improvement was
found at the urban site JSPAES, as E2 failed to improve the emission estimation of
local sources including transportation and residential activities. Compared to $NH_3$, the
modeling performance for inorganic aerosols was better for most cases, and the
differences between the simulated concentrations with E1 and E2 were clearly smaller,
at 7%, 3% and 12% (relative to E1) for $NH_4^+$, $SO_4^{2-}$, and $NO_3^-$, respectively.
Regarding the satellite-derived $NH_3$ column, application of E2 significantly corrected
the overestimation in vertical column density simulation for January and October with
E1, but did not improve the model performance for July. The $NH_3$ emissions might be
underestimated with the assumption of linear correlation between $NH_3$ volatilization
and soil pH for acidic soil, particularly in warm seasons. Three additional cases, i.e.,
40% abatement of $SO_2$, 40% abatement of $NO_X$, and 40% abatement of both species
were applied to test the sensitivity of $NH_3$ and inorganic aerosol concentrations to





precursor emissions. Under an $NH_3$-rich condition for most of YRD, estimation of
$SO_2$ emissions was detected to be more effective on simulation of secondary inorganic
aerosols compared to $NH_3$. Reduced $SO_2$ would restrain the formation of $(NH_4)_2SO_4$,
and thereby enhance the $NH_3$ concentrations. Besides the emissions, uncertainties
existed as well in the limitations of ground and satellite observation and incomplete
mechanism of gas to particle conversion applied in the model. To improve the air
quality more effectively and efficiently, $NH_3$ emissions should be substantially
controlled along with $SO_2$ and $NO_X$ in the future.

## 1.    Introduction

As the most important alkaline composition in the atmosphere, ammonia ($NH_3$)

exerts crucial influences on atmospheric chemistry and nitrogen cycle. $NH_3$
participates in chemical reactions with sulfur dioxide ($SO_2$) and nitrogen oxides
($NO_X$), and contributes to formation of secondary inorganic aerosols (SIA) including
sulfate ($SO_4^{2-}$), nitrate ($NO_3^-$), and ammonium ($NH_4^+$) and to thereby the elevated
concentrations of fine particulate matters (PM). In the developed regions in eastern
China, for example, SIA was observed to account for over 50% of $PM_{2.5}$ mass
concentrations (Yang et al., 2011; Zhang et al., 2012; Huang et al., 2014), and $NH_3$
emissions were estimated to contribute 8-11% of $PM_{2.5}$ (Wang et al., 2011). Recent
studies reported that existence of $NH_3$ could accelerate the heterogeneous oxidation of
$SO_2$ and thereby sulfate formation by neutralizing aerosol acidity (Wang et al., 2016;
Cheng et al., 2016; Paulot et al., 2017). Deposition of gaseous $NH_3$ and $NH_4^+$ aerosol
results in soil acidification and water eutrophication. Reduced nitrogen ($NH_3+NH_4^+$)
was monitored to contribute over 70% of total nitrogen deposition in China, revealing
the importance of $NH_3$ on ecosystem (Pan et al., 2012). Along with the improved
controls of $SO_2$ and $NO_X$ emissions, enhanced contribution of $NH_3$ emissions was
found to secondary aerosol formation and nitrogen deposition for recent years in
China (Liu et al., 2013; Fu et al., 2017; Pan et al., 2018).

Quantification of $NH_3$ sources helps better understanding its atmospheric and

ecosystem effects. In contrast to $SO_2$ and $NO_X$ that are largely from industrial plants,
$NH_3$ comes mainly from agricultural activities that are more difficult to track,
including livestock farming and fertilizer use, and relatively large uncertainty in $NH_3$
emission inventories existed. Given the intensive agriculture across the country,



various methods were developed to estimate China's $NH_3$ emissions at the national
level for last twenty years, but clear discrepancies existed between studies, as
summarized by Zhang et al. (2018). With meteorology, soil property, the method of
fertilizer application and different processes of manure management considered in
emission factor (emissions per unit level of activity) determination, in particular, the
national $NH_3$ emissions estimated by Peking University group (Huang et al., 2012;
Kang et al., 2016) was 39-46% smaller than those by Tsinghua University group
(Dong et al., 2010; Zhao et al., 2013). Emissions of certain sectors differed
significantly between various methods. For example, Zhao et al. (2013) and
Kurokawa et al. (2013) calculated China's $NH_3$ emissions from fertilizer use at 9.5-9.8
Tg, over three times of the estimation by Kang et al. (2016). With a fertilizer
modeling system that couples an air quality model and an agro-ecosystem model, Fu
et al. (2015) made an estimate at 3.0 Tg, similar with Kang et al. (2016). Besides the
annual emission level, discrepancies existed as well in the inter-annual trend of
emissions. Kang et al. (2016) estimated that the national $NH_3$ emissions reached peak
in 1996 and declined, while Zhang et al. (2017) and Kurokawa et al. (2013) expected
a continuous growth till 2008 and 2015, respectively. The growth in $NH_3$ emissions
got supported by satellite observation. Based on the measurement of Atmospheric
Infrared Sounder (AIRS), for example, Warner et al. (2017) suggested an annual
increasing rate of $NH_3$ concentrations at 2.3% from 2002 to 2016 in China, and it was
partly attributed to the elevated emissions from fertilizer use.

Although varied methods and data resulted in discrepancies between inventories

and big uncertainty in $NH_3$ emission estimation, very little attention has been paid to
those discrepancies and the underlying reasons. At regional scale, in particular,
inclusion of high-resolution information on meteorology and land use would
potentially improve the spatial and seasonal distribution of agricultural $NH_3$ emissions
in the inventory. Moreover, few studies were conducted to carefully evaluate $NH_3$
emission inventories incorporating air quality model and available ground and
satellite observations. One possible reason is the lack of sufficient ground observation
data on $NH_3$ and $NH_4^+$ aerosols open to public, as they are currently not regulated air
pollutants in China and thus not regularly monitored by the government. In addition,
uncertainty also existed in satellite observation on $NH_3$ columns and the retrieved data
needs further validation (van Damme et al., 2015). Without comparison of different
inventories in details and appropriate assessment based on model performance, the





limitations of current emission estimates and the future steps for inventory
improvement remained unclear.
In this study, therefore, we chose the Yangtze River Delta (YRD) region to
develop and evaluate high-resolution emission inventories of $NH_3$ with different
methods and data sources. Located in the eastern China, the YRD region contains the
city of Shanghai and the provinces of Jiangsu, Zhejiang and Anhui (see Figure 1 for
its location and prefectural cities), and is one of China's most developed and
heavy-polluted regions (Xiao et al., 2011; Cheng et al., 2013; Guo et al., 2017). It is
an important area of agriculture production, and was identified as a "$NH_3$-rich" region
regarding the SIA formation (Wang et al., 2011). We developed $NH_3$ emission
inventories for 2014 based on the constant emission factors (Method 1) and those
characterizing the agricultural processes (Method 2). The two inventories were
compared against each other to reveal the differences in spatial and seasonal patterns
of $NH_3$ emissions and their origins. Evaluation of the two inventories was further
conducted using a Models-3 Community Multi-scale Air Quality (CMAQ) system and
available observations from ground station and satellite. Environmental parameters
that might influence $NH_3$ simulation were identified through the model performance.
Finally, the effects of $SO_2$ and NOx emission estimates on $NH_3$ and $NH_4^+$ aerosol
simulation were evaluated through sensitivity analysis, and the policy implication of
air quality improvement were accordingly suggested.

## 2.   Data and methods

**2.1 Emission inventory based on constant emission factors (E1)**

The annual $NH_3$ emissions of the YRD region for 2014 were estimated with a
bottom-up method based on constant emission factors. The inventory contained eight
source categories, i.e., fertilizer application, livestock/poultry breeding, fuel
combustion, biomass burning, transportation, sewage/waste treatment, industrial
process, and human metabolization (see Table 1 for details). The emissions were
calculated by prefectural city with the Eq. 1:

$$E_{i,j} = \sum_j \left( AL_{i,j} \times EF_j \times 10^{-3} \right) \tag{1}$$

where $E$ is the emissions, metric ton (t); $i$ and $j$ indicate the prefectural city and source
type, respectively; $AL$ is the activity level, which indicated the amount of livestock,



the amount of used fertilizer, the fuel burned or the industrial production, depending
on the source type; and $EF$ is the emission factor, kg-$NH_3$/unit $AL$.
Activity data were mainly taken or estimated from official statistics at the
prefectural city (if available) or provincial level. For livestock/poultry breeding, the
year-end stock and slaughtered numbers were used respectively for animals with the
breeding cycle more and less than one year. If the city-level stock was unavailable, the
output of livestock products by prefectural city was applied as the scaling factor to
calculate the number from the provincial data. Table S1 in the supplement summarizes
the annual numbers of livestock and poultry by prefectural city in YRD. The amount
of fertilizer using by prefectural city and type was calculated as the product of sown
area of cropland and fertilizer rate per unit area of cropland. The sown area by crop
type was taken from city-level statistics, and the application rate by fertilizer type was
obtained at provincial level from a national investigation by NDRC (2015). The
detailed results of fertilizer activity data are summarized in Table S2 in the
supplement. As can be seen as well in the table, the aggregated amount of fertilizer
using by province were close to the provincial-level statistics, and the deviation
relevant to the official statistics was 2.3% for the whole YRD. The methods and data
sources for activity levels of other source categories were provided in our previous
studies (Zhou et al., 2017; Zhao et al., 2017; Yang and Zhao, 2019).
The $NH_3$ emission factors were obtained based on a thorough literature review
and summarized by source category in Table S3 in the supplement. The results from
domestic field measurements were preferentially selected. For sources without
suitable domestic measurements, the emission factors were also obtained from
previous inventories that shared similar studying period with this work. The values
from US and Europe, e.g., AP-42 database (USEPA, 2002) and the EMEP/EEA
guidebook (EEA, 2013), were adopted as well when above information was lacking.
The monthly distribution of emissions by source was taken from domestic
investigations in YRD (Li, 2012; Zhao et al., 2015; Zhou et al., 2017). For the purpose
of air quality modeling, the emissions by sector were allocated into a grid system with
a horizontal resolution at 9×9 km based on selected proxies. Those proxies included
the distribution of land use (for fertilization), density of total population (for human
metabolization and sewage/waste treatment) and rural population (for
livestock/poultry breeding and residential solid fuel burning), gross domestic product
(for industrial fuel combustion and processes), road net (for transportation), and the
satellite-derived fire points from Moderate Resolution Imaging Spectroradiometer (for
open biomass burning).
**2.2 The method characterizing the agricultural processes (Method 2)**
The emissions from fertilizer use and livestock/poultry breeding were corrected
or recalculated integrating the detailed regional information of soil, meteorology and
agricultural processes, as described below.
**2.2.1 Fertilizer use**
The growing seasons of crops affects the temporal distribution of fertilizer use
and thereby that of $NH_3$ emissions. We investigated the growing and farming cycles
by crop type in YRD from the regional farming database by the Ministry of
Agriculture (MOA, http://202.127.42.157/moazzys/nongshi.aspx) and other
publication (Zhang et al., 2009), and corrected the monthly amount of fertilizer using
by prefectural city and fertilizer type combining the information of farming season
and annual fertilizer using as given in Table S2.
Emission factors of fertilization were expected to be influenced by soil acidity,
temperature, and the fertilization rate. We assumed a near-linear correlation between
the soil pH and $NH_3$ volatilization rate (Huang et al., 2012), and calculated the
emission factors of two fertilization types (basal dressing and top dressing) with Eq. 2
and 3, respectively:
$$EF_{base} = [(a_{pH} \times pH + b_{pH}) + (T_{base} - T_0 - 273.15) \times k_T] \times CF_{rate} \times CF_{method} \qquad (2)$$
$$EF_{top} = [(a_{pH} \times pH + b_{pH}) + (T_{top} - T_0 - 273.15) \times k_T] \times CF_{rate} \qquad (3)$$
where $EF_{basal}$ and $EF_{top}$ are the emission factors for basal dressing and top dressing,
respectively; $a_{pH}$ and $b_{pH}$ are the corrected slope and intercept depending on soil pH;
$T_0$ and $k_T$ are the reference temperature and corrected slope depending on temperature,
respectively; $T_{basal}$ and $T_{top}$ are the temperature of basal dressing and top dressing,
respectively; and $CF_{rate}$ and $CF_{method}$ are the corrected factors for fertilization rate and
method, respectively.
The spatial distribution of soil pH at a horizontal resolution of 1×1km was
obtained from a world soil database by International Institute for Applied Systems and
Analysis                                                                            (IIASA,
http://webarchive.iiasa.ac.at/Research/LUC/External-World-soil-database/HTML/).
The correlation data between temperature and $NH_3$ volatilization rate were obtained
from EEA (2013). $T_{basal}$ and $T_{top}$ were determined combining the information of



farming season by MOA and the daily temperature data from European Centre for
Medium-Range Weather Forecasts (ECNWF,
http://apps.ecmwf.int/datasets/data/interim-full-daily/levtype=sfc/#userconsent#). All
the relevant data for emission factor correction were summarized in Table S4 in the
supplement. The corrected $NH_3$ volatilization rates of urea and ammonium
bicarbonate (ABC), the mostly applied two types of fertilizer over the YRD region,
were illustrated by season in Figure S1 in the supplement. Larger volatilization rates
were found in northern YRD for both fertilizer types, consistent with the distribution
of soil pH across the region. Taking urea as an example, the volatilization rates in
April and October were commonly smaller than the uniform value applied in E1 at
17.4%, while those in July were larger. This discrepancy came partly from the
consideration of fertilization types in E2. In April and October, basal dressing
fertilization was commonly applied at the soil depth of 15-20 centimeters, restraining
the $NH_3$ volatilization. In contrast, the relatively high temperature and top dressing
fertilization conducted in July elevated the $NH_3$ volatilization.
Through the methodology mentioned above, the gridded emission factors and
monthly activity levels were obtained to improve the spatial and temporal
distributions of $NH_3$ emissions from fertilization. Figure 2 compares the activity data
of fertilization between the two methods (E1 and E2), indicated by the relative
deviation (*RD*):
$$RD = (E_1 - E_2)/(E_1 + E_2)/2 \qquad (4)$$
In January and July, top dressing fertilization was conducted with limited crop types
like rape, corn and paddy rice, while considerable basal dressing fertilization was
investigated in April and October. Inclusion of those details in E2 resulted in smaller
estimates of fertilizer use in winter and summer but larger in spring and autumn
compared to E1.

### 2.2.2 Livestock/poultry breeding

In contrast to Method 1 that calculated the $NH_3$ emissions based on livestock
numbers and annual EFs, a mass-flow approach was applied in Method 2 considering
the nitrogen transformation at different stages of manure management (Beusen et al.,
2008; Huang et al., 2012; EEA, 2013). Commonly applied at global or national scale,
the approach calculated $NH_3$ emissions of manure management processes from a pool
of total ammoniacal nitrogen (TAN) for three main raising systems, as shown in


Figure S2 in the supplement. In YRD region, only intensive and free-range systems
were considered, and the TAN was calculated by livestock/poultry type based on the
breeding duration, the amount and nitrogen contents of urine/feces, and the mass
fraction of TAN. The parameters were taken from Yang (2008) and Huang et al (2012),
as summarized in Table S5 in the supplement. According to the nitrogen flow and
phase of manure management, the activity levels were then classified into seven
categories, including outdoor, housing (solid), housing (liquid), storage (solid),
storage (liquid), spreading (solid) and spreading (liquid). $NH_3$ emissions from
livestock are calculated as the product of TAN of each category and corresponding
emission factors. As provided in Table S6 in the supplement, the
temperature-dependant emission factors by stage/phase were taken from EEA (2013)
and Huang et al. (2012), and the gridded emission factors can then be derived over the
YRD region combining the meteorology data from ECNWF.

**2.3 Configuration of air quality modeling**
The Models-3 Community Multi-scale Air Quality (CMAQ) version 4.7.1 was
applied to evaluate the $NH_3$ emission inventories for YRD. CMAQ is a
three-dimensional Eulerian model designed for understanding the complex
interactions of atmospheric chemistry and physics (http://www.cmaq-model.org). The
model has been widely applied and tested in China (Qin et al., 2015; Zhou et al., 2017;
Zheng et al., 2019). As shown in Figure 1, two nested domains were applied with the
spatial resolutions of 27 and 9 km respectively, on a Lambert Conformal Conic
projection centered at ($110^o$E, $34^o$N). The mother domain (D1, $177\times127$ cells)
covered most parts of China, and the second domain (D2, $118\times121$ cells) covered the
whole YRD region. The two inventories of YRD $NH_3$ emissions developed in this
work were applied in D2. Emissions from other pollutants of anthropogenic origin in
D1 and D2 outside Jiangsu were obtained from the Multi resolution Emission
Inventory for China (MEIC, http://www.meicmodel.org/) with an original spatial
resolution of 0.25°×0.25°. Population density was applied to relocate MEIC to each
modeling domain. A high-resolution inventory that incorporates more information of
local emission sources was applied for Jiangsu (JS, Zhou et al., 2017). Both MEIC
and JS inventories are for 2012. The emissions for 2014 were obtained using a simple
scaling method based mainly on changes in activity levels (energy consumption and

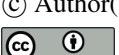



industrial production, etc) between the three years. Biogenic emission inventory was
from the Model Emissions of Gases and Aerosols from Nature (MEGAN, Sindelarova
et al., 2014), and the emission inventories of Cl, HCl and lightning $NO_X$ were from
the Global Emissions Initiative (GEIA, Price et al., 1997). Meteorological fields were
provided by the Weather Research and Forecasting Model (WRF) version 3.4, a
state-of-the-art atmospheric modeling system designed for both meteorological
research and numerical weather prediction (http://www2.mmm.ucar.edu/wrf/users/),
and the carbon bond gas-phase mechanism (CB05) and AERO5 aerosol module were
adopted. Other details on model configuration and parameters were given in Zhou et
al. (2017). The simulations were conducted for January, April, July and October to
represent four typical seasons in 2014. A 5-day spin-up period of each month was
used to minimize the influences of initial conditions in the simulations.
Using the observation data of US National Climate Data Center (NCDC) at 43
stations in YRD (see Figure 1 for the locations of the stations), the WRF modeling
performance was evaluated with statistical indicators including averages of
simulations and observations, bias, normalized mean bias (NMB), normalized mean
error (NME), root mean squared error (RMSE) and index of agreement (IOA). As can
be found in Table S7 in the Supplement, discrepancies between simulation and
observation met the criteria by Emery et al. (2001) for most cases, implying the
reliability of meteorological simulation. However, bigger errors were found for the
simulation of wind direction.
**2.4 Ground-based and satellite observations**
There were very limited continuous ground measurement data available for
ambient $NH_3$ and $NH_4^+$ aerosol in the YRD region 2014, particularly at rural/remote
sites that are more representative for the regional atmospheric environment. We
conducted on-line hourly measurements using the MARGA (Monitor for AeRosols
and Gases in ambient Air, ADI2080) at an urban site in the western downtown of
Nanjing (32.03°N, 118.44°E) from August 2014. The MARGA is a state-of-art
instrument which monitors near real-time water-soluble ions in aerosols and their
gaseous precursors (Lanciki, 2018), and it was able to capture rapid compositional
changes in $PM_{2.5}$ (Chen et al., 2017). The site was on the roof of the building of
Jiangsu Provincial Academy of Environmental Science (30 m above the ground)
surrounded by residential and commercial buildings and heavy traffic (JSPAES: Li et





al., 2015; Chen et al., 2019). The data of October 2014 were applied in this work to
evaluate the NH$_3$ inventories through air quality simulation. Besides, the hourly data
of online measurement with MARGA were available at a suburban site in Pudong,
Shanghai (SHPD) for April, July and October 2014 (unpublished data from Shanghai
Environmental Monitoring Center).

Regarding satellite observation, the daily NH$_3$ vertical column densities (VCDs)

measured through Infrared Atmospheric Sounding Interferometer (IASI) were
downloaded          from          ESPRI          data          center
([http://cds-espri.ipsl.upmc.fr/etherTypo/index.php?id=1700&L=1](http://cds-espri.ipsl.upmc.fr/etherTypo/index.php?id=1700&L=1)). We used the data
in the domain (114.2°E-124.1°E, 26.1°N-35.4°N) with a 9:30am equator local
crossing time to evaluate the NH$_3$ emissions. Only pixels with radiative cloud
fraction<25%, relative error <100% and absolute error<$5\times10^{15}$ molec/cm$^2$ were used
following the criteria of previous studies (van Damme et al., 2014; 2015). The
monthly average VCDs for January, April, July and October 2014 were calculated and
allocated into a grid system of 0.5° (longitude)×0.25° (latitude) using the Kriging
interpolation method, as shown in Figure 3.

## 3.   Results and discussions

### 3.1  Comparison between the two inventories

Table 2 summarizes the NH$_3$ emissions estimated with E1 and E2 by source

category and province for the YRD region in 2014. Agricultural activities (livestock
farming and fertilizer) were identified as the most important sources of NH$_3$, with the
fraction to total emissions ranged 74-84% in the two methods. Applying the constant
emission factors, E1 derived a total NH$_3$ emission estimate 60% larger than that by E2
that characterized the agricultural processes. In particular, emissions from agricultural
activities in E1 were calculated as twice of those in E2. At the national scale, similarly,
Dong et al. (2016) applied constant emission factors and estimated the total NH$_3$
emissions at 16.1 Tg for China, 64% larger than 9.8 Tg by Huang et al. (2012) with
the agricultural processes characterized. The clearly larger estimation by constant
emission factors was due mainly to the fact that most emission factor measurements
were conducted in hot seasons. Among the provinces, the fraction of Jiangsu to YRD
emissions was ranged 45-47% in the two methods, followed by Anhui around 37%.
Agricultural activities were relatively intensive in the two provinces: Jiangsu and



Anhui contributed 46% and 33% of the economic output of agriculture and
livestock/poultry farming in YRD region, and the collective fraction of fertilizer use
by the two provinces reached 84%. In contrast, agricultural activities were limited in
Shanghai and Zhejiang, with smaller emissions estimated in both inventories.
Monthly distribution of $NH_3$ emissions in the two inventories were illustrated in
Figure 4. Both inventories indicated relatively large emissions in summer (from June
to August), and elevated emissions were also found in March and September in E2,
The difference comes mainly from the effect of farming season on fertilization
process. For example, the top dressing fertilization for winter wheat was conducted
mostly during the seedling establishment and elongation stage in the following spring,
resulting in enhanced use of nitrogen fertilizer in March. Moreover, September was
the month with the highest temperature following summer in YRD 2014, and the
elevated $NH_3$ led to large emissions in E2. Compared to fertilizer use, less variation of
monthly emissions were found for livestock/poultry breeding, as very limited change
in livestock amount was detected in both inventories.
Illustrated in Figure 5 are the spatial distributions of emissions from fertilizer use,
livestock/poultry breeding and all categories in the two inventories. Both inventories
indicated the large emission intensities in northern Jiangsu (Xuzhou and Yancheng)
and northern Anhui (Fuyang, Bozhou and Suzhou) with abundant agricultural
production. Xuzhou and Yancheng collectively contributed 36%, 31% and 41% of the
provincial fertilizer use, agricultural economic product, and livestock/poultry farming
product in Jiangsu, respectively. Similarly, Fuyang, Bozhou and Suzhou collectively
contributed 36%, 36% and 35% of the provincial sown area, agricultural economic
product, and livestock/poultry farming product in Anhui, respectively.
The differences in spatial pattern between the two inventories were further
investigated for total and fertilizer use emissions by month, through the indicator $RD$
calculated with Eq. (4). As shown in Figure 6, larger $RD$ was found in northern
Jiangsu, northern Anhui, and eastern Zhejiang, while smaller in western Zhejiang. The
emissions in E1 were commonly larger than that in E2 across the YRD region for
January and April. In contrast, larger emissions in E2 were found in northern Jiangsu
(e.g., Xuzhou and Yancheng) and northern Anhui for July and October. The
discrepancy resulted from the combined effect of varied activity data and emission
factors as described in Section 2.2: top dressing fertilization and high temperature led
to enhanced volatilization rate and thereby emissions of $NH_3$ in E2, and the abundant





fertilizer use in the broad cropland in northern YRD region was the main reason for the high emissions in October.

Figure 7 compares the $NH_3$ emissions by province and source category in this work and other available downscaled national (MEIC) or provincial inventories in the YRD region. Results from other studies were commonly ranged between E1 and E2 for agriculture, the most important $NH_3$ source. With constant emission factors applied, the MEIC estimates were similar to those in E1. Most current provincial inventories made some corrections for emissions from fertilizer use or livestock/poultry breeding, but the local geographical and meteorological information were not always fully applied in the emission estimation. For example, Liu and Yao (2016) calculated the emissions from livestock/poultry breeding for Jiangsu based on TAN, but did not consider the impacts of varied monthly temperatures on the emissions. Zheng et al. (2016) calculated the agricultural $NH_3$ emissions for Anhui based on a national guideline of $NH_3$ emission inventory development (MEP, 2014), and ignored the impact of soil condition (e.g., pH) on $NH_3$ volatilization from fertilizer use.

**3.2 Evaluation of the inventories with transport modeling and ground observation**

Figures 8 illustrates the observed and simulated hourly concentrations for gaseous $NH_3$ and inorganic aerosol species ($NH_4^+$, $SO_4^{2-}$ and $NO_3^-$) in ambient particles for April, July and October at SHPD and October at JSPAES. The normalized mean biases (NMB) and normalized mean errors (NME) between observed and simulated concentrations, and the monthly average concentrations from observation and simulation are summarized in Table 3. In general, the model captured the temporal variation of $NH_3$ concentrations, and the simulated monthly average concentrations were close to the observed ones at both sites. The biggest discrepancy was found at SHPD for April, where the monthly average $NH_3$ was simulated 56% larger than observation with E1, and the smallest at JSPAES for October, where the simulated was 1.7% smaller than observation with E1. The simulated temporal variation, however, was much larger than the observation, leading to relatively large NME, particularly at SHPD for April. Clear difference was found for the simulation under two $NH_3$ inventories. In general, the average of simulated $NH_3$ concentrations at the two sites for available months was 27% smaller in E2 than that in E1 (note the

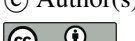



total NH$_3$ emissions in E2 was 38% smaller than that in E1 for the whole YRD region).
At SHPD site, application of E1 in CMAQ overestimated the NH$_3$ concentration,
indicated by the positive NMB values and the larger simulated concentrations than
observation. Such overestimation was clearly corrected when E2 was applied, and the
NMEs with E2 were substantially reduced as well, as shown in Table 3. The better
modeling performance implies the improved estimation and spatiotemporal
distribution of emissions. At JSPAES, air quality modeling with both inventories
underestimated the NH$_3$ concentrations, and the simulated monthly average
concentration with E1 was much closer to observation than that with E2. The close
NMEs between the two inventories indicated very limited improvement at the site, in
contrast to SHPD. Located in urban area, JSPAES might be largely affected by the
local sources like transportation and residential activities. NH$_3$ emissions of such
source categories, however, were not improved in E2.
To reduce the impact of highly uncertain hourly meteorology simulation and
emission data on air quality modeling, daily NH$_3$ concentrations derived from
simulation and observation were further compared for October at JSPAES. As
illustrated in Figure S3 in the supplement, better agreement between observation and
simulation was achieved for daily concentration than hourly, and the NMEs for E1
and E2 were reduced respectively from 56.9% and 53.7% to 37.0% and 32.5%,
respectively. Besides emission data, uncertainty in meteorology simulation also
contributed to the discrepancy between simulation and observation. For example, both
inventories overestimated the concentration on 7$^{th}$ October but underestimated that for
21$^{st}$-22$^{nd}$. In contrast to the southeasterly wind observed at ground meteorology station
in Nanjing, the simulated wind direction on 7$^{th}$ was from north, enhancing the NH$_3$
transport from Yancheng and Xuzhou in northern Jiangsu with intensive agricultural
activities and thereby emissions. On 21$^{st}$-22$^{nd}$, the underestimation NH$_3$ concentration
resulted largely from the overestimation in wind speed by WRF.
Compared to NH$_3$, the modeling performance for inorganic aerosols (NH$_4^+$, SO$_4^{2-}$,
and NO$_3^-$) is better for most cases, indicated by smaller NMEs in Table 3. Some
exceptions exist at SHPD for NH$_4^+$ and SO$_4^2$ in October and NO$_3^-$ in January.
Application of E2 reduced the NMEs and improved the simulation of NH$_4^+$ and SO$_4^2$
moderately, but there were no significant changes between the modeling results with
E1 and E2. The averages of simulated concentrations at the two sites for available
months was 7%, 3% and 12% smaller in E2 than those in E1 for NH$_4^+$, SO$_4^{2-}$, and





$NO_3^-$, respectively, and the differences were clearly smaller than that for $NH_3$ at 27%.
As large fraction of inorganic aerosols comes from secondary chemistry reaction, they
are more representative for the regional atmosphere condition other than the local
environment around the measurement site. Therefore, the air quality modeling at a
horizontal resolution at 9×9 km is expected to be able to better simulate the
concentrations for secondary inorganic aerosols than primary gaseous pollutants,
particularly when emissions from some local sources are not sufficiently quantified.
The simulated concentrations were commonly larger than observation for $NH_4^+$ and
$SO_4^{2-}$, particularly at SHPD in July and October. The uncertainty of model could be an
importance source of the discrepancy, as the recent reported mechanisms of gas to
particle conversion were not sufficiently applied in the CMAQ we used (Wang et al.,
2016; Cheng et al., 2016). In addition, positive or negative artifacts also existed in
ground observation with MARGA, resulting from the unexpected reaction between
acid gaseous pollutants and nitrate aerosol (Wei et al., 2015). From an emission
perspective, the overestimation was partly corrected when smaller $NH_3$ emissions in
E2 were applied instead of E1 in the model. Due to missing information on individual
industrial plants, moreover, the inventory we used in CMAQ failed to fully capture
the progress of emission control in YRD region and probably overestimated the $SO_2$
emissions (Zhang et al., 2019). The formation of sulfate ammonium aerosols could
then be enhanced through the irreversible reaction between $SO_2$ and $NH_3$. The process
simultaneously reduced the amount of $NH_3$ reacted with $HNO_3$, leading further to the
underestimation of nitrate aerosols. As shown in Table 3, application of E2 with less
$NH_3$ emissions than E1 could not improve the modeling performance of nitrate
aerosols. The impact of $SO_2$ and $NO_X$ emission on SIA modeling will be further
discussed in Section 3.4.
**3.3 Evaluation of the inventories with transport modeling and satellite**
**observation**
To be consistent with the local crossing time of IASI at 9:30am, the average of
simulated hourly $NH_3$ concentrations at 9:00 am and 10:00 am were applied to
calculate the $NH_3$ VCDs, using the following equations:
$$n_{NH3} = \sum_{k=1}^{23} m_k \times \Delta H_k \times 100 \tag{5}$$
$$\Delta H_k = H \times \ln\left(\frac{p_k}{p_{k+1}}\right) \tag{6}$$



where $n_{NH3}$ is the NH$_3$ VCDs from CMAQ model (molec./cm$^2$); $m_k$ is the simulated
NH$_3$ concentrations at vertical layer k in the CMAQ (molec./cm$^3$); $\triangle H$ is the height of
layer $k$ (m); $H$ represents the height when the pressure of atmosphere declines to 1/e
of the original value; and p is the air pressure. Figure 9 illustrates the simulated NH$_3$
VCDs with E1 and E2 for January, April, July, and October. Similar spatial patterns
are found with the two inventories, i.e., relatively large NH$_3$ VCDs were simulated
mostly in northern Jiangsu and northern Anhui province, consistent with the hotspot
of NH$_3$ emissions. The simulated NH$_3$ VCDs with E1 were 53% larger than those with
E2 across the whole YRD region, with the maximum and minimum monthly
difference calculated at 73% and 31% for April and October, respectively. The NMB,
NME, and spatial correlation coefficient (r) between observed and simulated VCDs,
and the monthly average VCDs from observation and simulation are summarized in
Table 4. Application of both inventories resulted in larger NH$_3$ VCDs than those from
satellite observation for January and October, while simulated VCDs for April and
July were smaller. Besides the uncertainty from monthly distribution of NH$_3$
emissions, the bias from WRF modeling on temperature might also contribute to the
discrepancy between simulated and observed VCDs. As shown in Table S7 in the
supplement, WRF overestimated the monthly temperature in January and October
with the NMBs calculated at 26.6% and 0.34%, and underestimated it in April and
July with the NMBs calculated at-1.62% and -2.51%. Compared to E1, application of
E2 significantly reduced the NMEs from 83.8% to 37.5% for January and largely
corrected the overestimation in VCD simulation for January and October. The
simulated VCDs were 4.3% larger and 1.4% smaller than observation for the two
months, respectively. The results implied satisfying agreement between the simulated
and observed VCDs over the YRD region. Improvement in NH$_3$ VCD simulation was
also found for April when E2 instead of E1 was applied in the air quality modeling,
with the NMEs reduced from 65.8% to 60.7%. For July, however, application of E2
did not improve the model performance, implying that current method in E2 could
possibly underestimate the NH$_3$ volatilization when the actual ambient temperature
was high. Besides emissions, the discrepancy could result from various factors
including the uncertainty in chemical mechanisms in CMAQ and environmental
condition. Errors from satellite retrieval could also contribute to the inconsistence
between simulation and observation. van Damme et al. (2014), for example, estimated
an error of 19% for the total NH$_3$ columns in Asia. As IASI does not provide the





averaging kernel moreover, uncertainty in NH$_3$ column retrieval could result from the
reduced sensitivity of satellite measurement towards the surface.
To further investigate the impact of soil pH on the emissions and thereby the
modeling performance on NH$_3$ VCDs, the soil in the YRD region was classified to
three types, acidic soil (pH≤6.5), neutral soil (6.5<pH≤7.5), and alkali soil (pH>7.5),
and the NMB and NME between the simulated and observed NH$_3$ VCDs were
calculated by soil type and month, as summarized in Table 5. For neutral and acidic
soil, application of E2 that considers the effect of farming season, geophysical
condition and manure management on NH$_3$ emission rates resulted in clearly smaller
NMEs than E1, implying the improvement in emission estimation. For acidic soil,
however, the NMBs were negative for all the months when E2 was applied, and the
NMEs were elevated compared to E1 except for January. Moreover, application of E2
resulted in negative NMBs for neutral and alkali soil in April and July as well. Those
results implied that E2 possibly underestimated the NH$_3$ emissions for acidic soil
particularly for warm seasons. With the correction of pH and temperature, the NH$_3$
volatilization rate from basal dressing fertilization was relatively low, indicating that
the current near-linear assumption between the soil pH and NH$_3$ volatilization rate
might not be appropriate for soil with low pH values for eastern China. As shown in
Figure S4 in the supplement, the measured NH$_3$ volatilization rates from urea and
ABC fertilizer use under relatively high soil pH (Zhang et al., 2002; Zhong et al.,
2006) were close to the estimated values in E2, but the measured results for acidic soil
were clearly larger than those in E2.
**3.4  Impacts of SO$_2$ and NOx emission estimates on simulated NH$_3$ and aerosols**
Besides meteorology condition, NH$_3$ emissions, and soil pH, the estimates of
SO$_2$ and NO$_X$ emissions could influence the NH$_3$ and SIA simulation as well. SO$_2$ can
be transformed to S (IV) through liquid phase reaction and then be oxidized to S (VI)
by O$_3$, or can be directly oxidized to H$_2$SO$_4$ by H$_2$O$_2$ or hydroxyl radical ( · OH).
HNO$_3$ can be formed through NO$_2$ oxidation by ·OH at daytime, or through hydrolysis
of N$_2$O$_5$ at aerosol surface at night. Normally NH$_3$ preferentially reacts with H$_2$SO$_4$
and relatively stable (NH$_4$)$_2$SO$_4$ is produced, while NH$_4$NO$_3$ could easily be
decomposed under high temperature or low humidity condition. Therefore, ambient
NH$_3$ concentrations and formation of NH$_4^+$ aerosols are influenced by the balance
between acidic (SO$_2$ and NO$_X$) and alkaline component (NH$_3$) emissions.


As described in Section 2.3, the SO$_2$ and NO$_X$ emissions for 2014 used in this
work were scaled from those for 2014 based on the changes in activity data. Ignorance
of emission control progress during 2012-2014 would probably result in
overestimation in emissions. The bias was evaluated through satellite observation. The
daily planetary boundary layer (PBL) SO$_2$ and tropospheric NO$_2$ VCDs were obtained
from the OMSO2 Level-3 product
(http://disc.sci.gsfc.nasa.gov/Aura/data-holdings/OMI/omso2e_v003.shtml) and the
POMINO Level-3 product from Ozone Monitoring Instrument (OMI), respectively.
As shown in Table S8 in the supplement, all the provinces in YRD had their SO$_2$ and
NO$_2$ VCDs substantially reduced during 2012-2014, and the VCDs declined by 48%
and 31% respectively for the whole region. From a recent unpublished emission study,
however, the SO$_2$ and NO$_X$ emissions were estimated to reduce only 16% and 8% in
the YRD region for the two years (personal communication with Cheng Huang from
Shanghai Research Academy of Environmental Science). It can be inferred that, the
overestimation of SO$_2$ emissions might enhance their reaction with NH$_3$ and thereby
the formation of (NH$_4$)$_2$SO$_4$ in the air quality modeling. The formation of NO$_3^-$, in
contrast, might be suppressed accordingly.

**3.4.1 Identification of NH$_3$-rich/-poor condition in YRD region**

To evaluate the non-linear relation between gaseous pollutant emissions (SO$_2$,
NO$_X$ and NH$_3$) and SIA concentrations for the YRD region, we follow Ansari and
Pandis (1998) and calculated the gas ratio (GR) based on the modeling results:

$$GR = \frac{([NH_3]+[NH_4^+]) - 2\times[SO_4^{2-}]}{[NO_3^-]+[HNO_3]} \quad (7)$$

where the species in the bracket indicated the simulated ambient concentration. A GR
smaller than 0 indicates a NH$_3$-poor condition, and the enhanced NH$_3$ emissions
strengthen the oxidation of SO$_2$ and lead to increased SO$_4^{2-}$ (Wang et al., 2011). A GR
larger than 1 indicates an NH$_3$-rich condition. Enhanced NH$_3$ emissions have smaller
effects on growth of SO$_4^{2-}$ concentrations, and elevated SO$_2$ emissions may accelerate
the formation of NO$_3^-$ aerosols, as the increased NH$_4^+$ and SO$_4^{2-}$ reduce the NH$_4$NO$_3$
capacity in the liquid phase (Seinfeld and Pandis, 2006). A neutral condition is judged
when GR is between 0 and 1.
Figure 10 illustrates the spatial distribution of simulated GR for YRD region by
month with E1 and E2 NH$_3$ inventories. Implied by the GR values larger than 1.0 for





most of the areas, the YRD region was identified under the $NH_3$-rich condition when
E1 was applied, except southwest Zhejiang. The judgment is consistent with previous
studies (Wang et al., 2011; Dong et al., 2014). With reduced $NH_3$ emissions in E2, The
areas under neutral or $NH_3$-poor condition expanded particularly for January and April.
The common $NH_3$-rich condition suggested potentially high sensitivity of SIA
formation to $SO_2$ and $NO_X$ emissions.

**3.4.2 Sensitivities of $NH_3$ and SIA to $SO_2$ and NOx changes**

Three more cases were developed to test the effect of $SO_2$ and $NO_X$ emission
estimates on $NH_3$ and SIA simulation: Cases 1, 2 and 3 assumed 40% abatement of
$SO_2$ emissions, 40% abatement of $NO_X$ emissions, and 40% abatement of emissions
both species, respectively. E1 was applied for $NH_3$ emission estimates in all the cases.
Table 6 summarizes the modeling performance at JSPAES and SHPD for different
cases in October. Clear changes in $NH_3$ and SIA simulation were found with varied
$SO_2$ emissions, while the effect of varied $NO_X$ emissions on air quality modeling was
much smaller. The bias between simulation and observation was partly corrected for
most cases, indicated by the smaller NMBs. Indicated by NMEs, however, the
modeling performance was less conclusive. NMEs for $NH_4^+$ and $SO_4^{2-}$ were reduced
for Cases 1 and 3, while increased NMEs were found for $NH_3$ and $NO_3^-$. Limitation in
the mechanisms of secondary aerosol formation can be an important reason for the
discrepancy. Under $NH_3$-rich condition, abatement of $SO_2$ emissions (Case 1) would
reduce the formation of $(NH_4)_2SO_4$, and thereby lead to growth of $NH_3$ concentrations.
This is consistent with the situation in North China Plain, another typical region
suffering aerosol pollution in China (Liu et al., 2018). The simulated $NH_3$ were 10.1%
and 11.7% larger than those in base case at JSPAES and SHPD, and the simulated SIA
($NH_4^+ + SO_4^{2-} + NO_3^-$) were 7.9% and 11.0% smaller than those in base case at JSPAES
and SHPD, respectively. Based on the modeling results in Table 3, as a comparison,
the simulated $NH_3$ concentrations with $NH_3$ emissions in E2 were calculated 23% and
28% smaller than those with E1 at JSPAES and SHPD for October, respectively, and
the analogue number for SIA concentrations were 5% at both sites. While the
estimation of $NH_3$ emissions played an important role on $NH_3$ simulation, the $SO_2$
estimation could be more effective on SIA simulation. Abatement of $NO_X$ emissions
(Case 2) was much less influential. Less NOx slightly weakened the competition of
SIA formation against $SO_2$, thus enhanced formation of $(NH_4)_2SO_4$ and decreased





NH$_3$ concentration were simulated at both sites, as shown in Table 6. When SO$_2$ and
NO$_X$ were simultaneously reduced in the model (Case 3), similar results were found
with Case 1, implying again that SO$_2$ could be a crucial species in SIA formation in
the YRD region. In addition, NO$_3^-$ aerosols were simulated to grow with the 40%
abatement of SO$_2$ and NO$_X$ emissions, and the benefits of SO$_2$ and NO$_X$ control were
partly weakened. To be more effective and efficient on regional air quality
improvement, therefore, the control of NH$_3$ emissions should be strengthened along
with other pollutants.

**4.    Conclusions**

We took the YRD region in eastern China as an example and developed two

inventories of NH$_3$ emissions for 2014 based on the constant emission factors (E1)
and those characterizing the agricultural processes (E2), respectively. Available
information from ground and satellite observation was applied to evaluate the
inventories through air quality modeling. Both inventories indicated that agricultural
activities (livestock farming and fertilizer use) were the most important sources of
NH$_3$, but clear differences exist in estimates and spatial and seasonal distribution of
NH$_3$ emissions. The total NH$_3$ emissions in E1 were estimated 60% larger than E2,
and the emissions from agriculture in E1 were double of E2. The information on
fertilization season and type from local investigation in E2 resulted in discrepancies in
monthly distributions of NH$_3$ emissions from E1, particularly in northern Yangtze
River Delta areas with abundant croplands. Differences in emission estimates lead to
varied NH$_3$ concentrations from CMAQ modeling. At the suburban SHPD site, the
overestimation in NH$_3$ concentration from CMAQ with E1 could be largely corrected
with E2, implying the improved estimation of NH$_3$ emissions by E2. At the urban site
JSPAES, however, very limited improvement was achieved when E1 was replaced by
E2 in the model, indicating that the emission estimation of local urban sources like
transportation and residential activities were not improved in E2. Compared to NH$_3$,
the modeling performance for inorganic aerosols is better for most cases, and
differences between the simulated concentrations with E1 and E2 were clearly smaller.
Application of E2 improved the simulation of NH$_4^+$ and SO$_4^{2-}$ moderately. For the
comparison with satellite-derived NH$_3$ column, application of E2 significantly
corrected the overestimation in VCD simulation for January and October with E1, but





did not improve the model performance for July. Combining the soil distribution, it
can be inferred that current method might underestimate the $NH_3$ volatilization for
acidic soil particularly in warm seasons. Judged by simulated GR, most of YRD
region was identified as an $NH_3$-rich condition except southwest Zhejiang. Through
sensitivity test in which $SO_2$ and NOx emissions were solely or simultaneously
reduced, estimation of $SO_2$ emissions was detected to be more effective on SIA
simulation compared to $NH_3$. Reduced $SO_2$ emissions would suppress the formation
of $(NH_4)_2SO_4$, and thereby lead to growth of $NH_3$ concentrations. The control of $NH_3$
emissions should be strengthened along with that of $SO_2$ and $NO_X$ for improving the
air quality more effectively and efficiently in the region.
This work is a tentative effort on $NH_3$ emission evaluation at regional scale.
Given the insufficient field measurements, the relation between environmental
conditions (e.g., temperature and soil pH) and $NH_3$ volatilization were not well
quantified, resulting in bias in emission estimation. Uncertainties come also from the
limitations in ground and satellite observation and incomplete mechanism of SIA
formation in current air quality model. For better understanding the role of $NH_3$
emissions in regional air quality, more measurements on both sources and ambient
concentrations are recommended in the future.

**Data availability**

The Multi-resolution Emission Inventory for China used in this study was
obtained at http://www.meicmodel.org/ (last access: 31 July 2019, Tsinghua
University, 2012). The high-resolution inventory for Jiangsu province was obtained in
Zhou et al. (2017) and can be accessed at http://www.airqualitynju.com/ (last access:
31 July 2019). The daily $NH_3$ VCDs measured through IASI was obtained from
ESPRI data center at http://cds-espri.ipsl.upmc.fr/etherTypo/index.php?id=1700&L=1
(last access: 31 July 2019). The two $NH_3$ emission inventories developed in this work
(E1 and E2) will be available with the publication of this paper at
http://airquality.nju.com.



**Author contributions**

YZ developed the strategy and methodology of the work and wrote the draft. MY ran the model and produced the figures. XH revised the method and provided useful comments. FC and JZ conducted ground observation of $NH_3$ and aerosols.

**Competing interests**

The authors declare that they have no conflict of interest.

**Acknowledgements**

This work was sponsored by Natural Science Foundation of China (91644220 and 41575142) and the National Key Research and Development Program of China (2017YFC0210106). We would like to acknowledge Qizhen Liu and Zhong Zou from Shanghai Environmental Monitoring Center and Yunhua Chang from Nanjing University of Information Science & Technology for the ground measurement data, Qiang Zhang from Tsinghua University and Cheng Huang from Shanghai Research Academy of Environmental Science for emission data, and Simon Whitburn from Université Libre de Bruxelles and Yuanhong Zhao from Peking University for satellite data processing.

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





**FIGURE CAPTIONS**

**Figure 1. Studying area and research domain. The blue dots and red triangles indicate the locations of 43 meteorological monitoring sites and 2 air quality monitoring sites, respectively, and the numbers of 1–41 represent the prefectural cities of Fuyang, Bozhou, Huaibei, Suzhou, Liuan, Hefei, Huainan, Bengbu, Chuzhou, Anqing, Tongling, Wuhu, Maanshan, Chizhou, Xuancheng, Huangshan, Xuzhou, Suqian, Lianyungang, Huaian, Yancheng, Yangzhou, Taizhou, Nanjing, Zhenjiang, Changzhou, Wuxi, Suzhou, Nantong, Huzhou, Jiaxing, Hangzhou, Shaoxing, Ningbo, Zhoushan, Quzhou, Jinhua, Taizhou, Lishui, Wenzhou, and Shanghai. The map data provided by Resource and Environment Data Cloud Platform are freely available for academic use (http://www.resdc.cn/data.aspx?DATAID=201).**

**Figure 2. Differences of fertilizer application between the two inventories in YRD ( $RD = (E_1 - E_2)/(E_1 + E_2)/2$ ).**

**Figure 3. The spatial distribution of monthly average of NH$_3$ vertical columns over YRD region from IASI satellite observation (Unit: $10^{15}$ mole./cm$^2$).**

**Figure 4. Monthly NH$_3$ emissions from fertilizer use and livestock farming in E1 and E2.**

**Figure 5. Spatial distribution of NH$_3$ emissions from fertilizer use, livestock farming and all categories in E1 and E2.**

**Figure 6. Differences of NH$_3$ emissions from fertilizer use and all categories between the two inventories ( $RD = (E_1 - E_2)/(E_1 + E_2)/2$ ).**

**Figure 7. Comparison between the estimated NH$_3$ emissions in this work and other studies by province and source category. "Others" indicate Fang et al. (2015), Liu and Yao (2016), Yu et al. (2016), and Zheng et al. (2016) for Shanghai, Jiangsu, Zhejiang, and Anhui, respectively.**

**Figure 8. The observed and simulated hourly NH$_3$ and SIA concentrations with the two inventories at JSPAES and SHPD sites**

**Figure 9. The NH$_3$ VCDs in the YRD region simulated with the two inventories by month.**

**Figure 10. The GR values in the YRD region simulated with the two inventories by month.**



**TABLES**

**Table 1. Anthropogenic NH$_3$ emission source categories**

| Category | Subcategory | Category | Subcategory |
|---|---|---|---|
| Fertilizer application | urea | Fuel combustion | industrial coal combustion |
|  | ammonium bicarbonate |  | industrial oil combustion |
|  | ammonium nitrate |  | industrial gas combustion |
|  | ammonium sulfate |  | domestic coal combustion |
|  | compound fertilizer |  | domestic oil combustion |
| Livestock Farming | beef cattle |  | domestic gas combustion |
|  | dairy cow | Biomass burning | straw burning |
|  | horse/donkey/mule |  | domestic firewood |
|  | sow |  | open |
|  | hog | Transportation | light duty gasoline vehicle |
|  | goat |  | heavy duty gasoline vehicle |
|  | sheep |  | light duty diesel vehicle |
|  | layer |  | heavy duty diesel vehicle |
|  | laying duck |  | motorcycle |
|  | broiler | Sewage and waste treatment | waste landfill |
|  | duck |  | waste incineration |
|  | goose |  | waste compost |
|  | rabbit |  | sewage treatment |
|  | cattle/buffalo | Industry sources | ammonium synthesis |
| Human being | human sweat |  | nitrogenous fertilizer |
|  | human breath |  | phosphate fertilizer |
|  | human excretion |  | coking |
|  | baby excretion |  |  |





**Table 2. Two anthropogenic NH$_3$ emission inventories in the YRD region in 2014 (Gg)**

| | Method | Livestock | Fertilizer | Chemical Industry | Biomass Burning | Waste Disposal | Traffic | Fuel Combustion | Human Beings | Total |
|---|---|---|---|---|---|---|---|---|---|---|
| Shanghai | E$_1$ | 14.9 | 11.9 | 0.1 | 0.3 | 5.0 | 1.9 | 5.1 | 5.5 | 44.5 |
| | E$_2$ | 6.5 | 9.0 | | | | | | | 33.2 |
| Jiangsu | E$_1$ | 340.8 | 357.4 | 14.1 | 29.1 | 6.0 | 8.6 | 5.2 | 30.8 | 791.9 |
| | E$_2$ | 145.6 | 257.1 | | | | | | | 496.5 |
| Zhejiang | E$_1$ | 115.7 | 93.8 | 2.4 | 10.6 | 6.9 | 7.7 | 4.7 | 28.3 | 270.1 |
| | E$_2$ | 37.4 | 49.3 | | | | | | | 147.2 |
| Anhui | E$_1$ | 241.5 | 314.9 | 14.7 | 35.9 | 2.8 | 3.3 | 7.3 | 37.7 | 658.2 |
| | E$_2$ | 102.3 | 185.9 | | | | | | | 389.9 |
| Total | E$_1$ | 712.7 | 778.0 | 31.2 | 75.9 | 20.7 | 21.6 | 22.3 | 102.2 | 1764.7 |
| | E$_2$ | 291.8 | 501.3 | | | | | | | 1067.0 |

**Table 3. Model performance statistics for concentrations of NH₃ and SIA from observation and CMAQ simulation with the two inventories at SHPD and JSPAES sites for available months.**

| | Indicator | SHPD_Apr | | SHPD_July | | SHPD_Oct | | JSPAES_Oct | |
|---|---|---|---|---|---|---|---|---|---|
| | | $E_1$ | $E_2$ | $E_1$ | $E_2$ | $E_1$ | $E_2$ | $E_1$ | $E_2$ |
| $NH_3$ | NMB(%) | 75.11 | 17.02 | 15.62 | -12.85 | 32.32 | -5.05 | 1.73 | -21.75 |
| | NME(%) | 141.08 | 103.59 | 88.72 | 78.00 | 98.36 | 76.25 | 56.94 | 53.68 |
| | Mean sim.(μg/m³) | 7.12 | 4.76 | 10.70 | 8.06 | 7.39 | 5.30 | 7.75 | 5.96 |
| | Mean obs.(μg/m³) | 4.58 | | 9.25 | | 5.58 | | 7.62 | |
| $NH_4^+$ | NMB(%) | -8.78 | -19.14 | 12.98 | 6.11 | 84.45 | 74.02 | 15.01 | 9.53 |
| | NME(%) | 40.07 | 40.78 | 64.26 | 61.76 | 100.23 | 91.69 | 42.27 | 40.7 |
| | Mean sim.(μg/m³) | 6.91 | 6.13 | 7.04 | 6.61 | 7.64 | 7.21 | 10.97 | 10.45 |
| | Mean obs.(μg/m³) | 7.58 | | 6.23 | | 4.14 | | 9.54 | |
| $SO_4^{2-}$ | NMB(%) | 24.08 | 14.05 | 50.86 | 46.84 | 91.92 | 90.41 | 14.38 | 12.53 |
| | NME(%) | 57.59 | 51.61 | 84.63 | 81.15 | 110.18 | 108.61 | 43.65 | 42.31 |
| | Mean sim.(μg/m³) | 14.75 | 13.56 | 14.60 | 14.21 | 14.53 | 14.41 | 15.5 | 15.25 |
| | Mean obs.(μg/m³) | 11.89 | | 9.68 | | 7.57 | | 13.56 | |
| $NO_3^-$ | NMB(%) | -59.13 | -65.20 | -78.10 | -94.24 | 29.46 | 12.60 | -6.55 | -14.18 |
| | NME(%) | 65.72 | 70.16 | 141.43 | 142.86 | 93.69 | 70.54 | 44.81 | 44.94 |
| | Mean sim.(μg/m³) | 4.93 | 4.19 | 5.39 | 4.64 | 7.32 | 6.37 | 17.53 | 16.1 |
| | Mean obs.(μg/m³) | 12.05 | | 9.01 | | 5.65 | | 18.76 | |

Note: obs. and sim. indicate the results from observation and simulation, respectively. The NMB and NME were calculated using following equations (P and O indicates the results from modeling prediction and observation, respectively):

$$NMB = \frac{\sum_{i=1}^{n}(P_i - O_i)}{\sum_{i=1}^{n} O_i} \times 100\% \text{;} \quad NME = \frac{\sum_{i=1}^{n}|P_i - O_i|}{\sum_{i=1}^{n} O_i} \times 100\%$$



**Table 4. Model performance statistics for NH$_3$ VCDs from IASI observation and CMAQ simulation using two inventories by month.**

|            | January |       | April |        | July  |        | October |       |
|------------|---------|-------|-------|--------|-------|--------|---------|-------|
|            | E$_1$   | E$_2$ | E$_1$ | E$_2$  | E$_1$ | E$_2$  | E$_1$   | E$_2$ |
| NMB(%)     | 77.02   | 4.29  | 28.49 | -59.12 | 12.19 | -34.12 | 29.46   | -1.77 |
| NME(%)     | 83.83   | 37.54 | 65.8  | 60.07  | 43.93 | 51.91  | 46.38   | 43.17 |
| r(P<0.01)  | 0.38    | 0.42  | 0.50  | 0.51   | 0.68  | 0.64   | 0.5     | 0.55  |
| Mean sim.  | 14.09   | 8.30  | 9.57  | 3.40   | 11.28 | 6.65   | 10.00   | 7.61  |
| IASI obs.  | 7.96    |       | 7.54  |        | 10.23 |        | 7.72    |       |





**Table 5 The NMBs and NMEs between simulated and observed NH₃ VCDs simulation by soil pH and month**

| pH | Statistics (%) | January | | April | | July | | October | |
|---|---|---|---|---|---|---|---|---|---|
| | | $E_1$ | $E_2$ | $E_1$ | $E_2$ | $E_1$ | $E_2$ | $E_1$ | $E_2$ |
| pH>7.5 | NMB | 114.88 | 28.04 | 81.41 | -38.99 | 43.3 | 4.24 | 67.99 | 46.95 |
| | NME | 117.8 | 49.27 | 89.23 | 44.38 | 56.11 | 48.13 | 71.49 | 57.44 |
| 7.5<=pH<6.5 | NMB | 92.82 | 9.19 | 44.6 | -54.14 | 39.27 | -10.78 | 44.01 | 11.13 |
| | NME | 95.83 | 34.16 | 64.13 | 54.7 | 52.52 | 45.54 | 52.54 | 37.69 |
| pH<=6.5 | NMB | 41.61 | -11.76 | 1.30 | -67.41 | -12.43 | -55.81 | 8.64 | -25.48 |
| | NME | 54.72 | 36.76 | 60.16 | 68.5 | 34.78 | 56.72 | 35.27 | 43.68 |





**Table 6 The modeling performance at JSPAES and SHPD in cases with different SO₂ and NOₓ emission estimates**

|  | Cases | JSPAES | | | SHPD | | |
|---|---|---|---|---|---|---|---|
|  |  | Increased/ Decreased % | NMB % | NME % | Increased/ Decreased % | NMB % | NME % |
| $NH_3$ | Base case |  | 1.73 | 56.94 |  | 32.32 | 98.36 |
|  | Case 1 | 10.14 | 11.09 | 59.02 | 11.67 | 47.54 | 102.68 |
|  | Case 2 | -1.17 | -0.59 | 57.85 | -0.83 | 29.51 | 96.93 |
|  | Case 3 | 8.48 | 9.29 | 59.64 | 11.12 | 44.92 | 100.94 |
| $NH_4^+$ | Base case |  | 15.01 | 42.27 |  | 84.45 | 100.23 |
|  | Case 1 | -8.67 | 5.19 | 39.24 | -10.99 | 62.53 | 84.93 |
|  | Case 2 | 1.87 | 17.55 | 45.40 | 1.40 | 87.40 | 102.37 |
|  | Case 3 | -6.95 | 7.33 | 41.85 | -10.36 | 65.69 | 86.27 |
| $SO_4^{2-}$ | Base case |  | 14.38 | 43.65 |  | 91.92 | 110.18 |
|  | Case 1 | -17.63 | -4.90 | 40.81 | -19.59 | 54.30 | 82.62 |
|  | Case 2 | 2.76 | 18.42 | 43.7 | 1.55 | 94.34 | 112.30 |
|  | Case 3 | -14.91 | -1.98 | 39.39 | -18.45 | 55.96 | 83.67 |
| $NO_3^-$ | Base case |  | -6.55 | 44.81 |  | 29.46 | 93.69 |
|  | Case 1 | 1.25 | -5.92 | 44.52 | 6.30 | 37.56 | 92.51 |
|  | Case 2 | 0.86 | -5.85 | 46.71 | -0.43 | 34.61 | 98.52 |
|  | Case 3 | 1.85 | -4.90 | 46.51 | 5.78 | 42.85 | 97.19 |



**Figure 1**

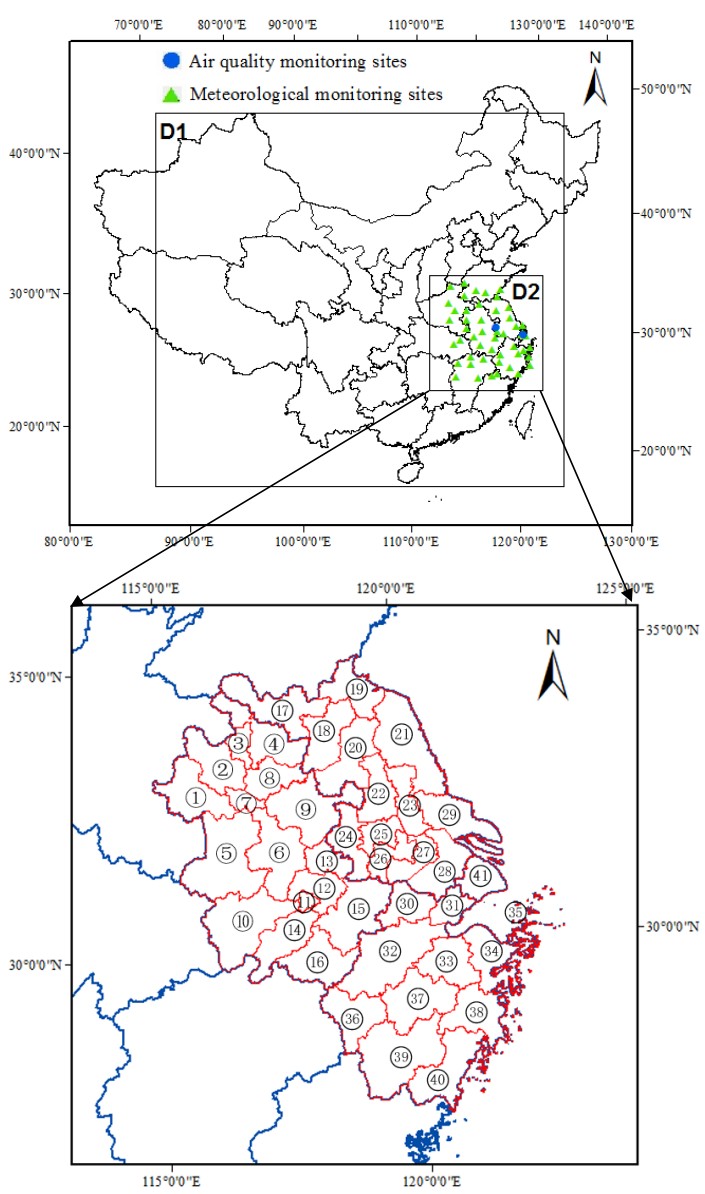





**Figure 2**

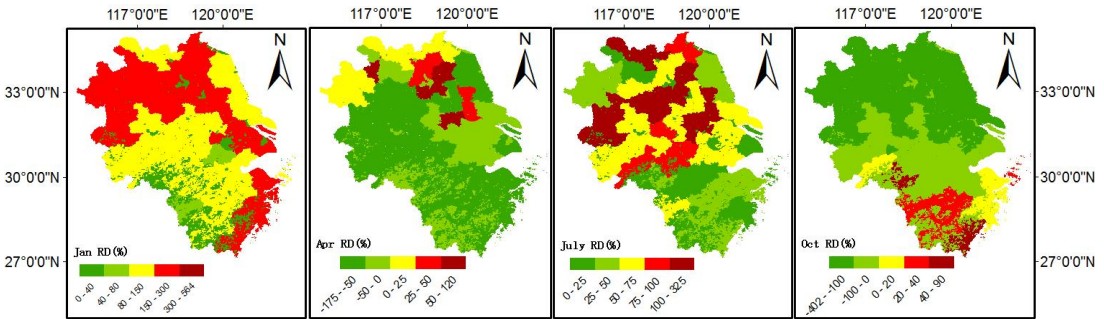



**Figure 3**

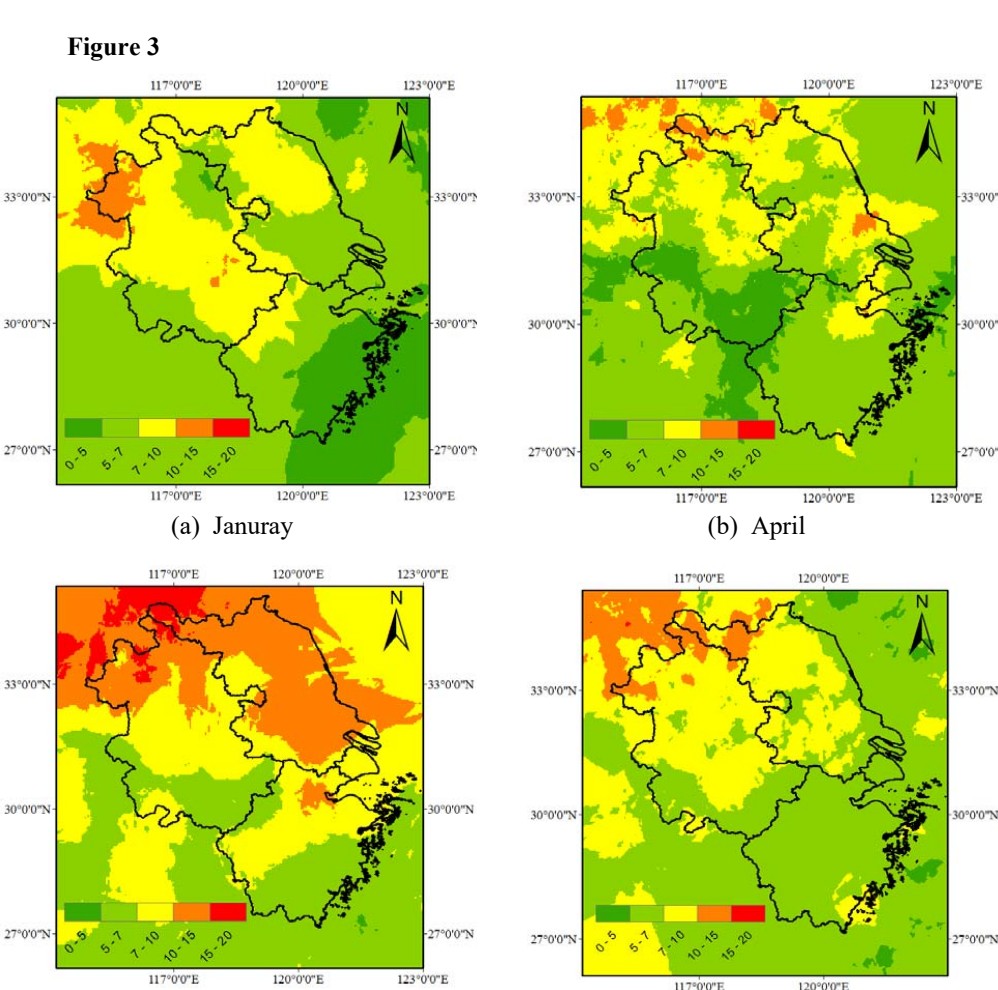

(a) Januray             (b) April

(c) July             (d) October





**Figure 4**

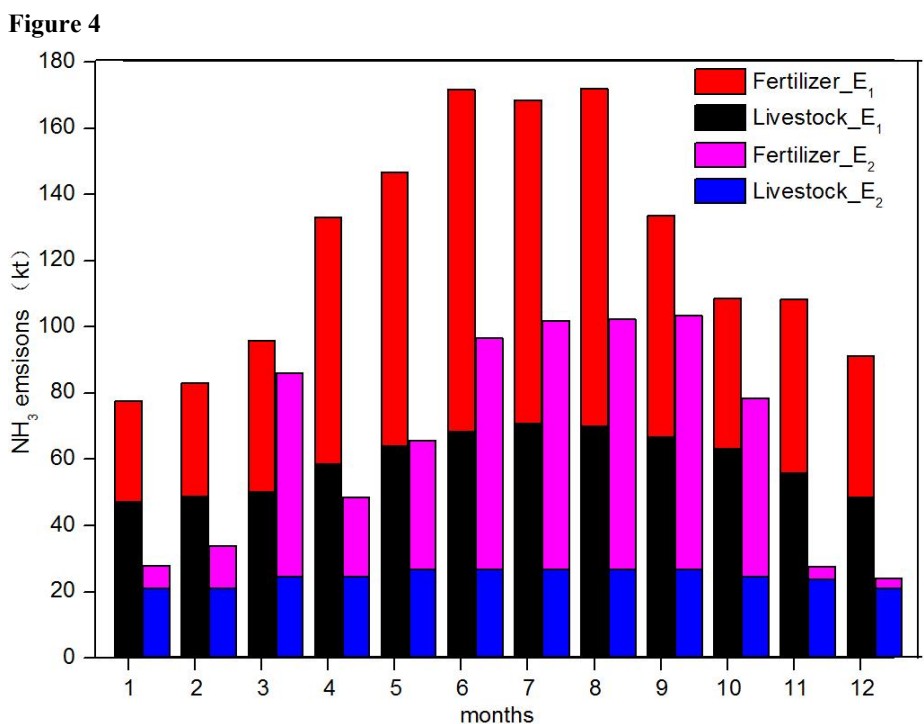



**Figure 5**

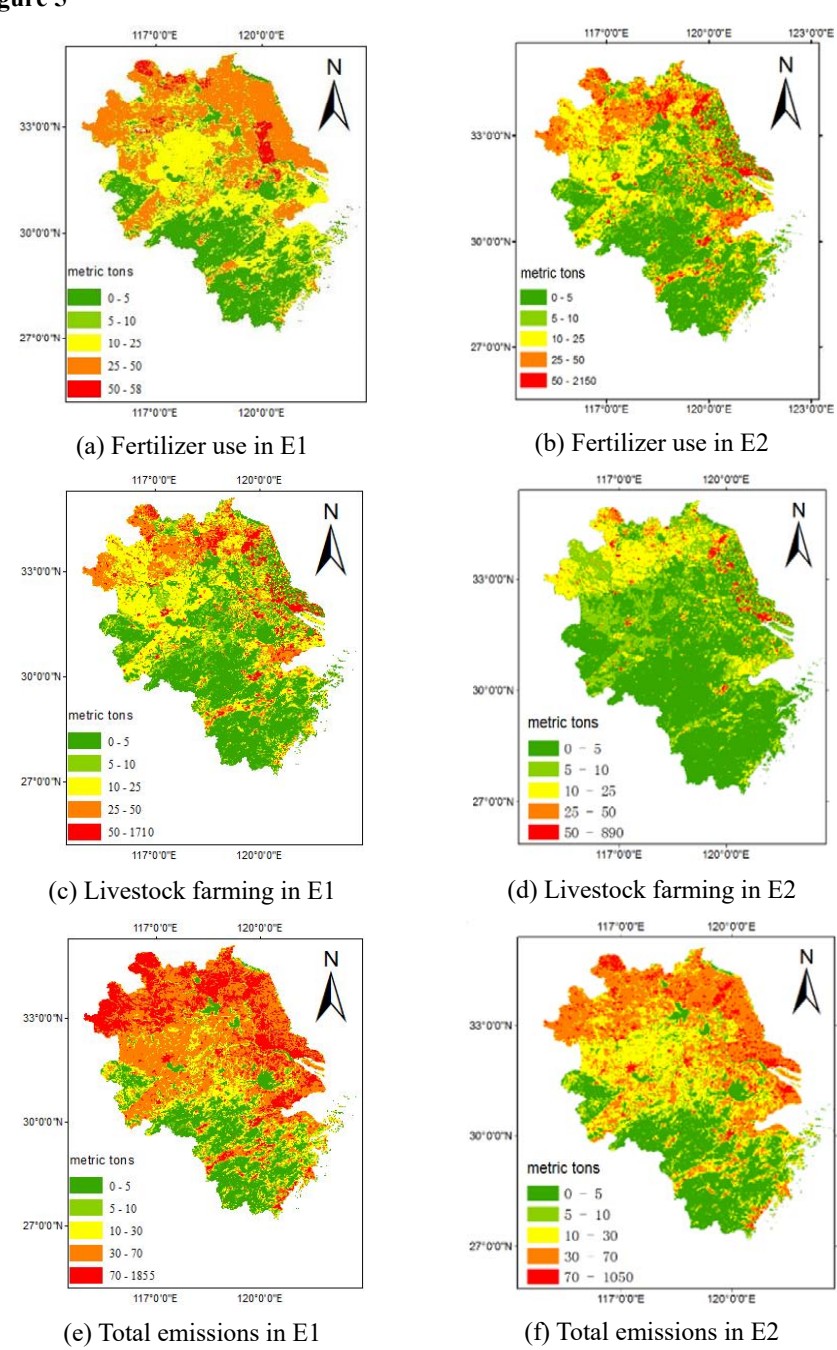



**Figure 6**

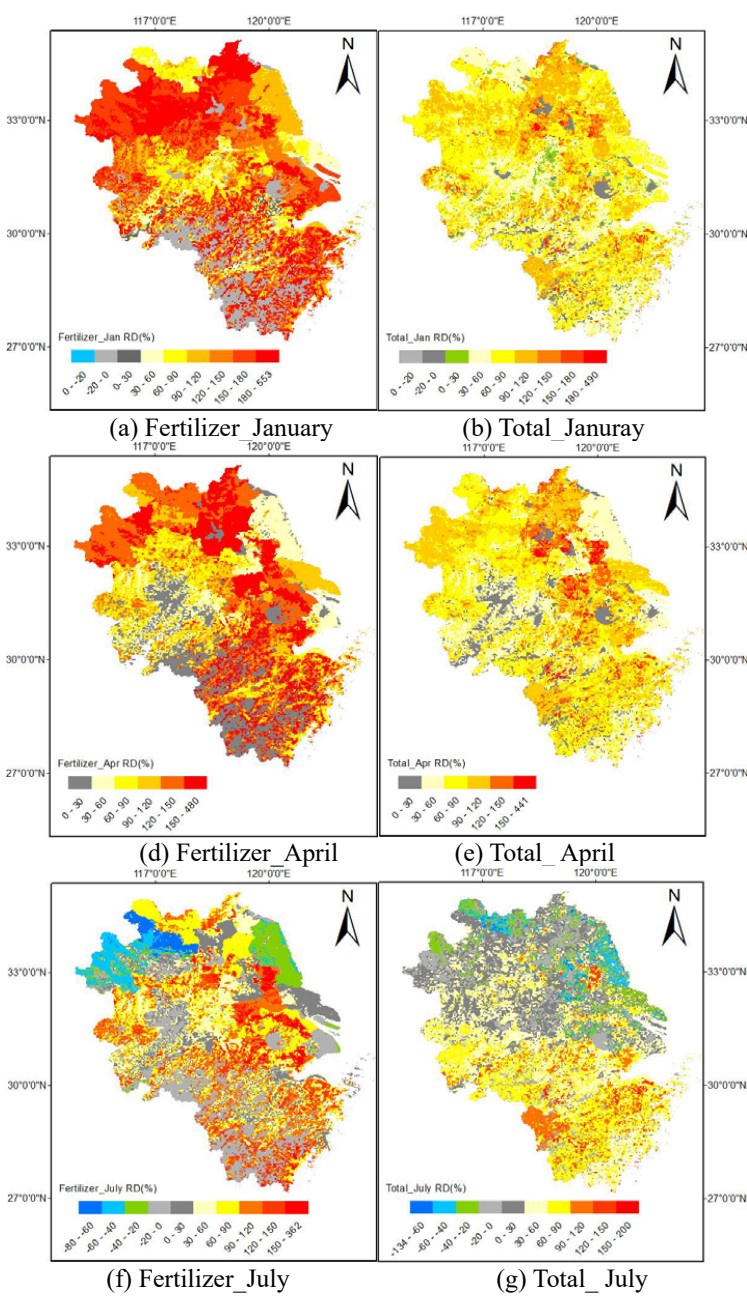



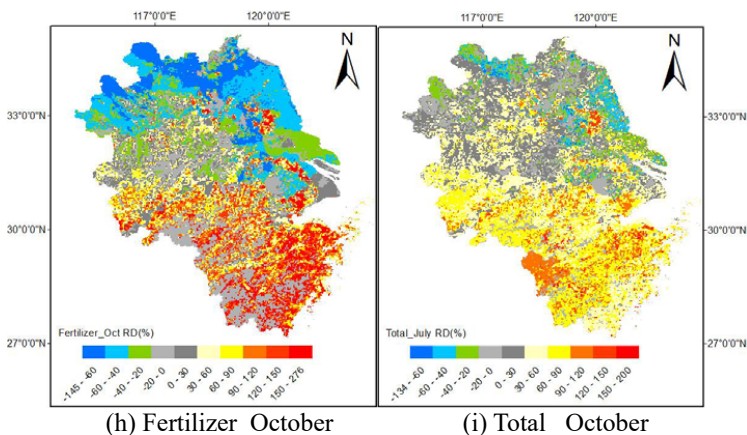

(h) Fertilizer_October          (i) Total_ October



**Figure 7**

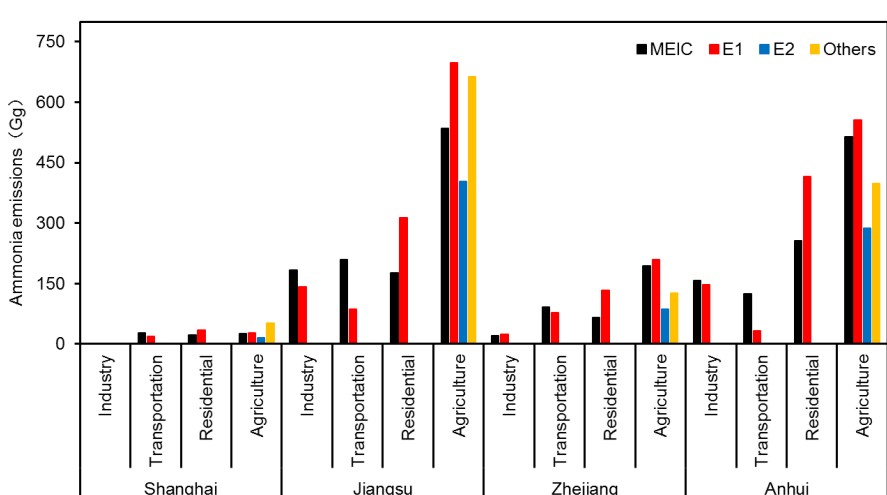

**Figure 8**



(a) SHPD_April

(b) SHPD_July



(c) SHPD_October

(d) JSPAES_October

**Figure 9**

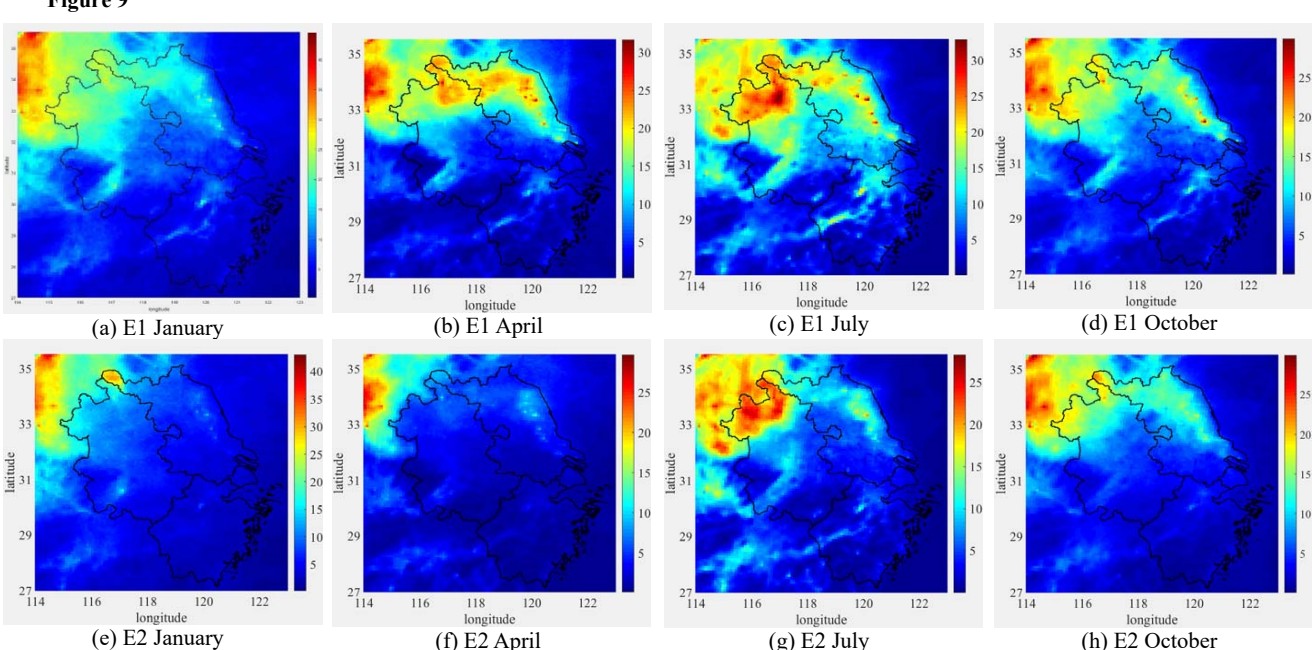

(a) E1 January     (b) E1 April     (c) E1 July     (d) E1 October

(e) E2 January     (f) E2 April     (g) E2 July     (h) E2 October



**Figure 10**

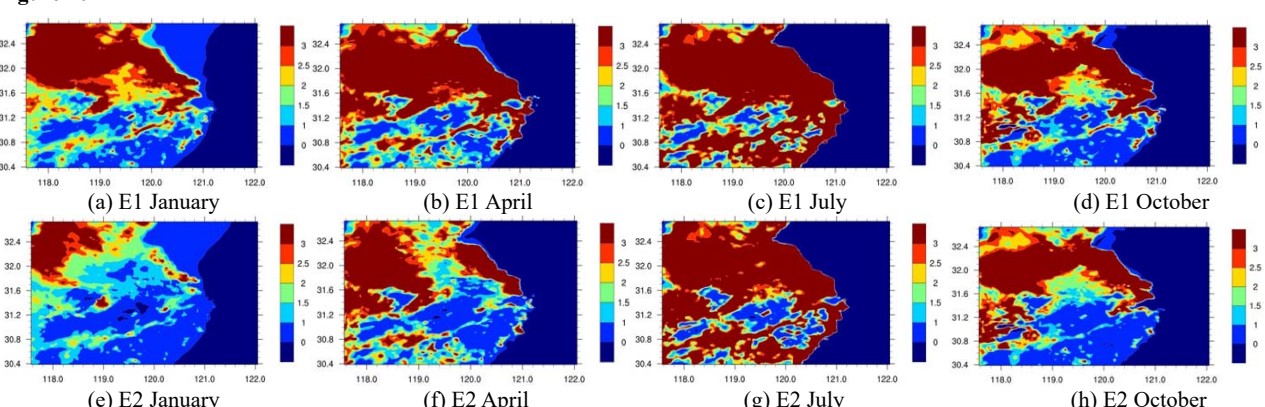

(a) E1 January     (b) E1 April     (c) E1 July     (d) E1 October

(e) E2 January     (f) E2 April     (g) E2 July     (h) E2 October