# Peer review of "Quantification and evaluation of atmospheric ammonia emissions with different methods: A case study for the Yangtze River Delta region, China"

_Atmospheric Chemistry and Physics, 2019_

## Referee Comment (RC1) · Anonymous Referee #2 · 20 Dec 2019

This paper compares and contrasts two methodologies for estimating emissions of NH3 in China, and illustrates the consequence through model simulations. The paper deals with an important subject, since the large uncertainties surrounding ammonia emissions need to be understood by modelers and policy experts.

The paper is generally well written, and generally sound, but I miss consideration of many of the factors omitted from the emission estimation procedure. This study basically used temperature, and agricultural statistics, to calculate emission factors (EFs). However, with respect to emissions from livestock/poultry, wind-speed is also a very

important factor (e.g. Gyldenkaerne et al., 2005, Skjoeth et al., 2011, Flechard et al, 2013). Many other factors should also impact NH3 emissions, such as radiation, rainfall (and other precipitation), leaf-wetness, atmospheric stability, large uncertainties in the so-called Gamma factors, or bi-directional exchange in general (Bash et al, 2013, Flechard et al., 2013, Massad et al, 2010, Wichink Kruit et al., 2012).

Consideration of such factors might also help to explain some if the model discrepancies outlined in Section 3, and should at least be considered before trying to explain all such discrepancies in terms of temperature and a few selected variables only.

The authors use meteorology from ECMWF for their emissions, but why not the WRF model, since that is obviously available and is used for their CMAQ runs?

The equations used are generally clearly written out, although it isn't always clear where they are coming from. For example, is it correct that equations 2 & 3 are a mixture of methods from Huang et al 2012 and EEA 2013? On the other hand, I read in various sections of EEA 2013 that temperature functions could not be provided (e.g. chap. 3.D crop production and agricultural soils) If from EEA, then it would also be good to cite the scientific papers underlying the EEA guidelines, and to be more specific as to which sections of EEA are being cited (it is a monster document).

Some other points:

P2. The abstract is rather long, and should be shortened for clarity.

P3, L67. NH3 is said to react with NOx, but NOx usually means NO+NO2. I think the authors mean HNO3?

P3, L78-81. The sentence is a little unclear. Clarify.

P4, L112. Methods of including meteorology in NH3 emissions have been around for some time and should be mentioned, e.g. Gyldenkaerne et al., 2005, Skjoeth et al., 2011, , Wichink Kruit et al., 2012, Bash et al., 2013.

P5, L148. Another source of human-related NH3 emissions is pets. As shown in e.g. Sutton et al 1995, 2000, human pets can be as significant as human metabolism with regard to NH3 emissions.

P6, L168. Using should be used.

P7, L187. Give reference for radiometer

P7, L202. The study of Huang et al 2012 uses a linear relationship between pH and EF. Why is the relation here said to be near-linear?

P7. What is the time-resolution of the EF calculations?

P8, L232. Surely fertilizer application at 15-20cm affects the pH of the soil; doesn't this affect the assumptions made when using global pH data from IIASA?

P9. The basic references of the CMAQ model should be given, not just a web-address.

P10. Which version of MEGAN was used? Did you use data provided by Sindelarova, or did you use the MEGAN model itself? If the latter, a Guenther et al ref would seem more

P10. Again, give reference to the model developers - this time for WRF.

P11. The Lanciki 2018 reference for MARGA is missing.

P15. The citation of Wei et al (2015) is in Chinese, and thus not helpful for most authors. This instrument has been around for many years, and the artifacts documented elsewhere. Please find some citations in English for the problems mentioned.

P28, Use molecule not "mole.", to avoid confusion with the mole unit.

P31. Table 3. Correlation coefficients should be added, and the time-resolution of the statistics mentioned.

P31 cont. for all Tables make it clear if statistics are calculated from hourly, daily or monthly values.

There are small English misses throughout, for example with regard to singular or plural, or omission of the definite article (the).

References

Bash, J. O., Cooter, E. J., Dennis, R. L., Walker, J. T. & Pleim, J. E. Evaluation of a regional air-quality model with bidirectional NH_3 exchange coupled to an agroecosystem model Biogeosciences, 2013, 10, 1635-1645

Flechard, C. R., Massad, R.-S., Loubet, B., Personne, E., Simpson, D., Bash, J. O., Cooter, E. J., Nemitz, E. & Sutton, M. A. Advances in understanding, models and parameterizations of biosphere-atmosphere ammonia exchange Biogeosciences, 2013, 10, 5183-5225

Gyldenkaerne, S., Skjøth, C., Hertel, O. & Ellermann, T. A dynamical ammonia emission parameterization for use in air pollution models J. Geophys. Res., 2005, 110

Skjoeth, C. A., Geels, C., Berge, H., Gyldenkaerne, S., Fagerli, H., Ellermann, T., Frohn, L. M., Christensen, J., Hansen, K. M., Hansen, K. & Hertel, O. Spatial and temporal variations in ammonia emissions - a freely accessible model code for Europe Atmos. Chem. Physics, 2011, 11, 5221-5236

Sutton, M. A., Dragosits, U., Tang, Y. S. & Fowler, D. Ammonia emissions from non-agricultural sources in the UK Atmos. Environ., 2000, 34, 855-869

Sutton, M., Place, C., Eager, M., Fowler, D. & Smith, R. Assessment of the magnitude of ammonia emissions in the United-Kingdom Atmos. Environ., 1995, 29, 1393-1411

Wichink Kruit, R. J., Schaap, M., Sauter, F. J., van Zanten, M. C. & van Pul, W. A. J. Modeling the distribution of ammonia across Europe including bi-directional surface-atmosphere exchange Biogeosciences, 2012, 9, 5261-5277

---

## Referee Comment (RC2) · Anonymous Referee #3 · 20 Jan 2020

The manuscript develops and presents two gridded NH3 emission inventories, one based on emission factors from the literature and a second with more process information. The two are compared against one other, as well as to two ground sites. CMAQ output was also be compared against satellite columns. This is a good exploration of what is known about NH3 emission patterns in the heavily populated Yangtze River Delta region. That said, without well understanding the methods E1 and E2, it was difficult to fully review manuscript.

Main comment:

[Figure]

Emission inventories with general emission factors or more detailed process have always been used, so at first read I am not sure why this is considered as case study of the methodology versus something like "Quantification and evaluation of atmospheric ammonia emissions for the Yangtze River Delta region, China".

The exact methods used for E1 and E2 are fairly confusing. The "constant emission factors" method that is referenced throughout are actually based on annual emission factors, with a monthly and spatial allocation schemes given on L179-L188. This needs to be clearer early on in the manuscript. Also, to confirm, neither allocation affects the total yearly emission? Are the activity factors different in E1 than what are used in E2?

Sect 2.2.1 about E2: please check each use of 'corrected' to make sure it is clear what/how/why something is being corrected. Specifically, L198 why does the fertilizer use need to be corrected? Where do the relationships in Table S2 come from?

Specific technical/style:

L206 EFbase -> EFbasal and Tbase -> Tbasal

L206-207 Are Tbasal and T0 in different units? Otherwise, 273.15 wouldn't be needed

L213 'method' -> 'application method'? (if I'm guessing correctly). What are the possible methods?

L347-L349 where is this shown about the EFs being from hot seasons?

L518-L519 Please reword. IASI is an instrument, so it cannot 'provide' an averaging kernel

Figure 1 caption: "Studying area and research domain" -> aren't study area and research domain the same?

Figure 3 and Figure 6: 'Januray" -> "January"

Figure 4: emissions misspelled in the y-axis label Figure 4: Suggest giving fertilizer and

livestock consistent colors, then keeping E1 as solid fill but E2 as hatched for easier reading

Figure 6: colorscales could have greater consistency

Figure 9: the subplots should have a consistent axis font size

Figure 10; from caption, shouldn't axis limits be same as Figure 9? Also, helpful to add the border lines like in Figure 9 so one is orientated where they are looking

Figure S4: there is one main cluster of data along the black line, but why is there seem to also be a second one? Also, what is the significance of the red dots, which do not fit well especially for the ABC panel?

---

## Author Comment (AC1) · 28 Feb 2020

Manuscript No.: acp-2019-689

Title: Quantification and evaluation of atmospheric ammonia emissions with different methods: A case study for the Yangtze River Delta region, China

Authors: Yu Zhao, Mengchen Yuan, Xin Huang, Feng Chen, Jie Zhang

We thank very much for the valuable comments and suggestions from reviewer #2,

which help us improve our manuscript significantly. The comments were carefully considered and revisions have been made in response to suggestions. Following is our point-by-point responses to the comments and corresponding revisionsãĂĆ

0. This paper compares and contrasts two methodologies for estimating emissions of NH3 in China, and illustrates the consequence through model simulations. The paper deals with an important subject, since the large uncertainties surrounding ammonia emissions need to be understood by modelers and policy experts. The paper is generally well written, and generally sound.

Response and revisions:

We appreciate the reviewer's positive remarks on the importance of the work.

1. I miss consideration of many of the factors omitted from the emission estimation procedure. This study basically used temperature, and agricultural statistics, to calculate emission factors (EFs). However, with respect to emissions from livestock/poultry, wind-speed is also a very important factor (e.g. Gyldenkaerne et al., 2005, Skjoeth et al., 2011, Flechard et al, 2013). Many other factors should also impact NH3 emissions, such as radiation, rainfall (and other precipitation), leaf-wetness, atmospheric stability, large uncertainties in the so-called Gamma factors, or bi-directional exchange in general (Bash et al, 2013, Flechard et al., 2013, Massad et al, 2010, Wichink Kruit et al., 2012). Consideration of such factors might also help to explain some if the model discrepancies outlined in Section 3, and should at least be considered before trying to explain all such discrepancies in terms of temperature and a few selected variables only

Response and revisions:

We thank and agree the reviewer's important comment. In this work, we mainly compared the magnitude and the spatial and temporal distribution of the YRD NH3 emissions estimated with two different methodologies, and evaluated the two inventories

through air quality modeling based on available satellite and ground observation within the region. Compared to E1, in particular, E2 included the impacts of the growing and farming cycles, soil properties (pH) and selected meteorological factor (temperature) on NH3 emissions for fertilizer using sector, and those of manure management processes and ambient temperature for livestock/poultry breeding. Besides the parameters we are concerned with, however, some other factors and processes also play important roles on atmosphere-land exchange of NH3, as pointed by the reviewer. Those factors/processes that were not considered in this work include given meteorological factors (e.g., wind speed, precipitation and leaf surface wetness), surface layer turbulence, air and surface heterogeneous-phase chemistry, and plant physiological conditions (Flechard et al, 2013). In general, those factors/processes could be integrated in the bi-directional surface–atmosphere exchange module coupled in the air quality modeling, and improved estimation of NH3 flux (emissions and depositions) were expected. The modeling system with the bi-directional NH3 exchange were reported to be able to reduce the biases and error in simulation of NHx (NH3 +NH4+) wet deposition and ambient aerosol concentrations for both US and Europe (Bash et al, 2013; Wichink Kruit et al., 2012). Limited studies on the bi-directional NH3 exchange were found for China (e.g., Fu et al., 2015). Out of the scope of current work, we did not focus on the bi-directional NH3 exchange module and did not include the module for emission evaluation and comparison. We agree with the reviewer that the ignorance of given parameters/process in the estimation could potentially further explain the discrepancy between the simulation and observation. A more comprehensive evaluation and comparison in NH3 emission inventories was thus suggested in the future, including the bi-directional NH3 exchange and the top-down constraint with inversed modeling. We have discussed this limitation and added relevant literatures in lines 561-580, page 18 in the revised manuscript.

2. The authors use meteorology from ECMWF for their emissions, but why not the WRF model, since that is obviously available and is used for their CMAQ runs?

Response and revisions:

We thank the reviewer's comment. We do not have very specific reason for using the ECMWF instead of WRF. When calculating the emissions, the underlying data open to the public were preferentially selected. ECMWF provided daily average data that satisfied our need of emission estimation and they were open to the public, thus we selected the dataset.

3. The equations used are generally clearly written out, although it isn't always clear where they are coming from. For example, is it correct that equations 2 & 3 are a mixture of methods from Huang et al 2012 and EEA 2013? On the other hand, I read in various sections of EEA 2013 that temperature functions could not be provided (e.g. chap. 3.D crop production and agricultural soils) If from EEA, then it would also be good to cite the scientific papers underlying the EEA guidelines, and to be more specific as to which sections of EEA are being cited (it is a monster document).

Response and revisions:

We appreciate the reviewer's comment. The specific EEA guidelines (EEA 2013a; 2013b; 2009) were provided in the revised manuscript. For Eq. 2 & 3, in particular, the linear relationships between NH3 volatilization rate and temperature/soil pH were described in Chap. 4.D crop production and agricultural soils of EEA (2009)/Huang et al. (2012), and we specified them respectively in lines 227-228, page 8 and lines 211-212, page 7 in the revised manuscript.

4. Some other points:

P2. The abstract is rather long, and should be shortened for clarity.

Response and revisions:

We thank the reviewer's comment and the abstract was shortened.

P3, L67. NH3 is said to react with NOx, but NOx usually means NO+NO2. I think the

authors mean HNO3?

Response and revisions:

We thank the reviewer's reminder and it is corrected as nitric acid (HNO3) in the revised manuscript.

P3, L78-81. The sentence is a little unclear. Clarify.

Response and revisions:

We thank the reviewer's comment. We mean that SO2 and NOX emissions have gradually decreased due to improved control, thus the NH3 emissions was found to play a greater role on the secondary aerosol formation and nitrogen deposition, compared to previous years. The sentence is rewritten in lines 72-76, page 3 in the revised manuscript: Recently the SO2 and NOX emissions have gradually decreased due to implementation of various pollution control measures in China, thus NH3 emissions were found to play a greater role on secondary aerosol formation and nitrogen deposition compared to previous years.

P4, L112. Methods of including meteorology in NH3 emissions have been around for some time and should be mentioned, e.g. Gyldenkaerne et al., 2005, Skjoeth et al., 2011, Wichink Kruit et al., 2012, Bash et al., 2013.

Response and revisions:

We thank and agree the reviewer's comment. We have added the relevant papers and description in lines 110-111, page 4 in the revised manuscript.

P5, L148. Another source of human-related NH3 emissions is pets. As shown in e.g. Sutton et al 1995, 2000, human pets can be as significant as human metabolism with regard to NH3 emissions.

Response and revisions:

We thank and agree the reviewer's comment. Due to lack of detailed statistic, we did not include pet emissions in current NH3 inventories. Given the relatively small fraction in total emissions (e.g., less than 2% for United Kingdom by Sutton et al.), we believe that the uncertainty was limited. We have added the explanation in lines 149-152, page 5 in the revised manuscript.

P6, L168. Using should be used.

Response and revisions:

We thank the reviewer's reminder and it is corrected in the revised manuscript.

P7, L187. Give reference for radiometer

Response and revisions:

We thank the reviewer's reminder and the reference for radiometer is given in the revised manuscript (Davies et al., 2009).

P7, L202. The study of Huang et al 2012 uses a linear relationship between pH and EF. Why is the relation here said to be near-linear?

Response and revisions:

We thank the reviewer's reminder and it is corrected as linear in the revised manuscript.

P7. What is the time-resolution of the EF calculations?

Response and revisions:

The time-resolution of EF calculation is monthly. In the method, the fertilization method (top or basal dressing) was month-dependent, and monthly average temperature was applied for the EF calculation. We have added the information in lines 212-213, page 7 and line 220, page 8 in the revised manuscript.

P8, L232. Surely fertilizer application at 15-20cm affects the pH of the soil; doesn't this affect the assumptions made when using global pH data from IIASA?

Response and revisions:

We thank and agree the reviewer's comment. Previous domestic experimental studies in China (e.g, Zhong et al., 2006) indicated that the fertilizer application would increase the soil pH, particularly for the acidic soils. Bias thus existed in soil pH from the global database, without considering the detailed schedule and method of fertilizer application. As the quantitative relation between the fertilizer application and soil pH was still lacking at the regional scale in China, we ignored the interaction between the fertilizer application and soil pH in Eqs.(2). We acknowledged the limitation and added the explanation in lines 243-248, page 8 in the revised manuscript.

P9. The basic references of the CMAQ model should be given, not just a web-address.

Response and revisions:

We thank the reviewer's reminder and the basic operational guidance of CMAQ by University of North Carolina was provided in the revised manuscript (UNC, 2010).

P10. Which version of MEGAN was used? Did you use data provided by Sindelarova, or did you use the MEGAN model itself? If the latter, a Guenther et al ref would seem more

Response and revisions:

We thank the reviewer's comment. We used the MEGAN 2.1. The literature (Guenther et al., 2012) has been added in the revised manuscript.

P10. Again, give reference to the model developers - this time for WRF.

Response and revisions:

We thank the reviewer's reminder and the reference of WRF is provided in the revised manuscript (Skamarock et al., 2008).

P11. The Lanciki 2018 reference for MARGA is missing.

Response and revisions:

We thank the reviewer's reminder and the information of Lanciki (2018) is provided in the revised manuscript.

P15. The citation of Wei et al (2015) is in Chinese, and thus not helpful for most authors. This instrument has been around for many years, and the artifacts documented elsewhere. Please find some citations in English for the problems mentioned.

Response and revisions:

We thank the reviewer's comment and provided English papers for the problem in lines 484-485, page 16 in the revised manuscript. (Chen et al., 2017; Schaap et al., 2011; Stieger et al., 2018).

P28, Use molecule not "mole.", to avoid confusion with the mole unit.

Response and revisions:

We thank the reviewer's reminder and molecule is used in the revised manuscript.

P31. Table 3. Correlation coefficients should be added, and the time-resolution of the statistics mentioned.

Response and revisions:

We thank the reviewer's comment. The correlation coefficients between the observation and simulation were added in the revised Table 3, and the time-resolution of the statics was hourly, as mentioned in the revised caption of the table.

P31 cont. for all Tables make it clear if statistics are calculated from hourly, daily or monthly values.

Response and revisions:

We thank the reviewer's reminder. The statistics in Tables 3 and 6 were calculated based on the hourly values, and those in Tables 4 and 5 were from the daily values (the

value of one hour (9:30am for satellite observation and the average of 9:00am-10:00am for simulation) per day). The information has been added in the revised captions of Tables 3-6.

There are small English misses throughout, for example with regard to singular or plural, or omission of the definite article (the).

Response and revisions:

We thank the reviewer's comment and the grammar errors are corrected in the revised manuscript.

References

Bash, J. O., Cooter, E. J., Dennis, R. L., Walker, J. T., Pleim, J. E.: Evaluation of a regional air-quality model with bidirectional NH3 exchange coupled to an agroecosystem model, Biogeosciences, 10, 1635-1645, 2013.

Chen, X., Walker, J. T., Geron, C.: Chromatography related performance of theMonitor for AeRosols and GAses in ambient air (MARGA): laboratory and field-based evaluation, Atmos. Meas. Tech, 10, 3893-3908, 2017.

Davies, D. K., Ilavajhala, S., Wong, M. M., and Justice, C. O.: Fire Information for Resource Management System: Archiving and Distributing MODIS Active Fire Data, IEEE Geosci. Remote Sens., 47, 72-79, 2009.

European Environment Agency (EEA): EMEP/CORINAIR Air Pollutant Emission Inventory Guidebook-2013, 3.B Manure management, available at: https://www.eea.europa.eu/publications/emep-eea-guidebook-2013/part-b-sectoral-guidance-chapters/4-agriculture/3-b-manure-management/view (last access: 25 Feb 2020), 2013a.

European Environment Agency (EEA): EMEP/CORINAIR Air Pollutant Emission Inventory Guidebook-2013, 3.D Crop production and agricultural soils, available at: https://www.eea.europa.eu/publications/emep-eea-guidebook-2013/part-b-sectoral-guidance-chapters/4-agriculture/3-d-crop-production/view (last access: 25 Feb 2020), 2013b.

European Environment Agency (EEA): EMEP/CORINAIR Air Pollutant Emission Inventory Guidebook-2009, 4.D Crop production and agricultural soils, available at: https://www.eea.europa.eu/publications/emep-eea-emission-inventory-guidebook-2009/part-b-sectoral-guidance-chapters/4-agriculture/4-d/4-d-crop-production-and-agricultural-soils.pdf/view (last access: 25 Feb 2020), 2009.

Flechard, C. R., Massad, R.-S., Loubet, B., Personne, E., Simpson, D., Bash, J. O., Cooter, E. J., Nemitz, E., Sutton, M. A.: Advances in understanding, models and parameterizations of biosphere-atmosphere ammonia exchange, Biogeosciences, 10, 5183-5225, 2013.

Fu, X., Wang, S., Ran, L., Pleim, J. E., Cooter, E., Bash, J. O., Benson, V., Hao, J.: Estimating NH3 emissions from agricultural fertilizer application in China using the bi-directional CMAQ model coupled to an agro-ecosystem model. Atmos. Chem. Phys., 15, 6637-6649, 2015

Guenther, A. B., Jiang, X., Heald, C. L., Sakulyanontvittaya, T., Duhl, T., Emmons, L. K., Wang, X.: The Model of Emissions of Gases and Aerosols from Nature version 2.1 (MEGAN2.1): an extended and updated framework for modeling biogenic emissions. Geosci. Model Dev., 5, 1471-1492, 2012.

Lanciki, A.: 2060 MARGA Monitor for AeRosols and Gases in ambient Air. Metrohm Process Analytics, Switzerland, available at: https://www.metrohm.com/en/products/process-analyzers/applikon-marga/ (last access: 10 Feb 2020), 2018.

Schaap, M., Otjes, R. P., Weijers, E. P.: Illustrating the benefit of using hourly monitoring data on secondary inorganic aerosol and its precursors for model evaluation. Atmos.

Chem. Phys., 11, 11041–11053, 2011

Skamarock, W. C., Klemp, J. B., Dudhia, J., Gill, D. O., Barker, D. M., Duda, M. G., Huang, X.-Y., Wang, W., Powers, J. G. A Description of the Advanced Research WRF Version 3. NCAR Tech. Note NCAR/TN-475+STR, 113 pp.doi:10.5065/D68S4MVH, 2008.

Stieger, B., Spindler, G., Fahlbusch, B. Muller, K., Gruner, A., Poulain, L., Thoni, L., Seitler, E., Wallasch, M., Herrmann, H.: Measurements of PM10 ions and trace gases with the online system MARGA at the research station Melpitz in Germany - A five-year study. J. Atmos. Chem., 75, 33-70, 2018.

University of North Carolina at Chapel Hill (UNC): Operational Guidance for the Community Multiscale Air Quality (CMAQ) Modeling System Version 4.7.1 (June 2010 Release), available at http://www.cmaq-model.org (last access: 10 Feb 2020), 2010.

Wichink Kruit, R. J., Schaap, M., Sauter, F. J., van Zanten, M. C., van Pul, W. A. J.: Modeling the distribution of ammonia across Europe including bi-directional surface atmosphere exchange, Biogeosciences, 9, 5261-5277, 2012.

Zhong, N., Zeng, Q., Zhang, L., Liao, B., Zhou, X., Jiang, J.: Effects of acidity and alkalinity on urea transformation in soil, Chinese Journal of Soil Science, 37, 1123-1128, 2006 (in Chinese).
* * *

---

## Author Comment (AC2) · 28 Feb 2020

Manuscript No.: acp-2019-689

Title: Quantification and evaluation of atmospheric ammonia emissions with different methods: A case study for the Yangtze River Delta region, China

Authors: Yu Zhao, Mengchen Yuan, Xin Huang, Feng Chen, Jie Zhang

We thank very much for the valuable comments and suggestions from reviewer #3,

which help us improve our manuscript significantly. The comments were carefully considered and revisions have been made in response to suggestions. Following is our point-by-point responses to the comments and corresponding revisions.

0. The manuscript develops and presents two gridded NH3 emission inventories, one based on emission factors from the literature and a second with more process information. The two are compared against one other, as well as to two ground sites. CMAQ output was also be compared against satellite columns. This is a good exploration of what is known about NH3 emission patterns in the heavily populated Yangtze River Delta region. That said, without well understanding the methods E1 and E2, it was difficult to fully review manuscript.

Response and revisions:

We appreciate the reviewer's positive remarks on the importance of the work.

1. Emission inventories with general emission factors or more detailed process have always been used, so at first read I am not sure why this is considered as case study of the methodology versus something like "Quantification and evaluation of atmospheric ammonia emissions for the Yangtze River Delta region, China". The exact methods used for E1 and E2 are fairly confusing. The "constant emission factors" method that is referenced throughout are actually based on annual emission factors, with a monthly and spatial allocation schemes given on L179-L188. This needs to be clearer early on in the manuscript. Also, to confirm, neither allocation affects the total yearly emission? Are the activity factors different in E1 than what are used in E2?

Response and revisions:

We appreciate the reviewer's important comment and acknowledged that some descriptions on the principles of the two methods were unclear in the original manuscript. The reviewer was correct for E1. It was developed based on the constant annual emission factors at the prefectural city level (as most of the activity data could be obtained at

the prefectural city level). Spatial and monthly allocations of emissions were then conducted, without changes in total yearly emissions. Following the reviewer's suggestion, we mentioned this at the beginning of Section 2.1 (lines 143-145, page 5 in the revised manuscript). It is a relatively quick and simple method, based on the previous understanding of NH3 emissions at regional scale (both the emission factor and temporal distribution). The effects of actual environmental conditions and agricultural activities on emission rate were not considered at a high temporal and spatial resolution.

In E2, the method for fertilizer application and livestock/poultry breeding (the main source categories of NH3) was improved. In particular, the emission factors were developed at the monthly resolution, integrating the effects of soil, meteorology and agricultural processes. Therefore the method did not only change the temporal pattern of emissions but also the magnitude of the annual emissions, as the emission factors developed in this method varied from the ones applied in E1, which were directly taken from previous studies.

It should also be noted that the annual activity data were the same for the two inventories at the prefectural city level, although the monthly distributions were different. We have clearly stated the relation between the activity data in the two inventories in lines 196-197, page 7 in the revised manuscript. Please also see our response to Question 2 of the reviewer.

2. Sect 2.2.1 about E2: please check each use of 'corrected' to make sure it is clear what/how/why something is being corrected. Specifically, L198 why does the fertilizer use need to be corrected? Where do the relationships in Table S2 come from?

Response and revisions:

We appreciate the reviewer's comment and acknowledged some "corrected" were confusing. In E2, the emissions from fertilizer use and livestock/poultry breeding were recalculated with a different method from E1. In general, it is not a correction of E1. Therefore we deleted unnecessary "corrected" in the methodology section of E2 (Section 2.2). Regarding the word "corrected" fertilizer as pointed by the reviewer, in particular, we actually means that the monthly fertilizer used was estimated in E2 combining the information of investigated farming cycles. The annual total fertilizer was the same as E1. We also added an example of early-season rice for better understanding the method. The relevant texts have been revised in lines 199-209, page 7 in the revised manuscript.

Regarding Table S2, we clearly stated that the annual total amount of fertilizer used were the same by prefecture city and type in the two inventories in lines 196-197, page 7 in the revised manuscript. The method of estimating the annual amount of fertilizer by prefecture city and type in Table S2 was described in lines in lines 165-171, page 6 in the revised manuscript.

3. Specific technical/style: L206 EFbase -> EFbasal and Tbase -> Tbasal

Response and revisions:

We thank the reviewer's reminder and terms are corrected in the revised manuscript.

L206-207 Are Tbasal and T0 in different units? Otherwise, 273.15 wouldn't be needed

Response and revisions:

We thank the reviewer's reminder and 273.15 was deleted in the revised manuscript.

L213 'method' -> 'application method'? (if I'm guessing correctly). What are the possible methods?

Response and revisions:

We thank the reviewer's reminder and it is revised as application method (basal dressing).

L347-L349 where is this shown about the EFs being from hot seasons?

Response and revisions:

We appreciate the reviewer's comment. Most of the measurements on emission factor of ammonia from fertilizer application were conducted in summer or late spring (Cai et al., 2002; Huo et al., 2015; Su et al., 2006), especially those using micrometeorological method. It is expectable since that the basal dressing of single-season rice and maize as well as top dressing of wheat are usually conducted in late spring or summer. However, the crop rotation varies a lot in China, and part of the nitrogen-containing fertilizer actually is not applied in hot seasons. Emission estimation based on those emission factors may thus overestimate the emission intensity of ammonia (Huo et al., 2015; Wang et al., 2011; Zhang et al., 2010). We have provided relevant literatures and added the above discussion in lines 361-369, page 12 in the revised manuscript.

L518-L519 Please reword. IASI is an instrument, so it cannot 'provide' an averaging kernel.

Response and revisions:

We thank the reviewer's reminder and the sentence is rewritten in lines 537-538, page 17 in the revised manuscript:

As the ESPRI product of NH3 VCDs we applied in the study does not provide the averaging kernel. . .

Figure 1 caption: "Studying area and research domain" -> aren't study area and research domain the same?

Response and revisions:

We thank the reviewer's reminder, and they are the same. We delete the "Studying area" in the figure caption.

Figure 3 and Figure 6: 'Januray' -> "January"

Response and revisions:

We thank the reviewer's reminder and the errors are corrected.

Figure 4: emissions misspelled in the y-axis label Figure 4: Suggest giving fertilizer and livestock consistent colors, then keeping E1 as solid fill but E2 as hatched for easier reading

Response and revisions:

We thank the reviewer's reminder and the figure is improved as required.

Figure 6: colorscales could have greater consistency

Response and revisions:

We thank the reviewer's reminder and the figure is improved as required.

Figure 9: the subplots should have a consistent axis font size

Response and revisions:

We thank the reviewer's reminder and the same font size is applied in all the subplots.

Figure 10; from caption, shouldn't axis limits be same as Figure 9? Also, helpful to add the border lines like in Figure 9 so one is orientated where they are looking

Response and revisions:

We thank the reviewer's reminder and revised the axis limits. The border lines have also been added in the revised Figure 10.

Figure S4: there is one main cluster of data along the black line, but why is there seem to also be a second one? Also, what is the significance of the red dots, which do not fit well especially for the ABC panel?

Response and revisions:

The black line is obtained through linear regression based on all the blue dots (including the "second cluster" mentioned by the reviewer).

Different from blue dots that are calculated for all the grids within the research domain

of this study, the red dots are taken from available field measurement studies, as we mentioned in the figure caption. The gap between them, in particular at lower soil pH, explained the possible uncertainty in current method, i.e., the current linear assumption between the soil pH and NH3 volatilization rate might not be appropriate for soil with low pH values for eastern China. We discuss it in lines 553-560, page 18 in the revised manuscript.

References

Cai, G. X., Chen, D. L., Ding, H., Pacholski, A., Fan, X. H., Zhu, Z. L.: Nitrogen losses from fertilizers applied to maize, wheat and rice in the North China Plain. Nutr. Cycl. Agroecosys., 63, 187-195, 2002.

Huo, Q., Cai, X., Kang, L., Zhang, H., Song, Y., Zhu, T.: Estimating ammonia emissions from a winter wheat cropland in North China Plain with field experiments and inverse dispersion modeling, Atmos. Environ., 104, 1-10, 2015.

Su, F., Huang, B., Ding, X., Gao, Z., Chen, X., Zhang, F., Kogge, M., Römheld, V.: Ammonia volatilization of different nitrogen fertilizer types, Soils, 38, 682-686, 2006 (in Chinese).

Wang, S., Xing, J., Jang, C., Zhu, Y., Fu, J. S., Hao, J.: Impact assessment of ammonia emissions on inorganic aerosols in East China using response surface modeling technique. Environ. Sci. Technol., 45, 9293-9300, 2011.

Zhang, Y., Dore, A. J., Ma, L., Liu, X., Ma, W., Cape, J. N., Zhang, F.: Agricultural ammonia emissions inventory and spatial distribution in the North China Plain. Environ. Pollut., 158, 490-501, 2010.

---

## Author Response (AR1)

**Main revisions and response to reviewers' comments**

Manuscript No.: acp-2019-689

Title: Quantification and evaluation of atmospheric ammonia emissions with different methods: A case study for the Yangtze River Delta region, China

Authors: Yu Zhao, Mengchen Yuan, Xin Huang, Feng Chen, Jie Zhang

We thank very much for the valuable comments and suggestions from the two reviewers, which help us improve our manuscript significantly. The comments were carefully considered and revisions have been made in response to suggestions. Following is our point-by-point responses to the comments and corresponding revisions.

**Reviewer #2**

*0. This paper compares and contrasts two methodologies for estimating emissions of NH$_3$ in China, and illustrates the consequence through model simulations. The paper deals with an important subject, since the large uncertainties surrounding ammonia emissions need to be understood by modelers and policy experts. The paper is generally well written, and generally sound.*

**Response and revisions:**

We appreciate the reviewer's positive remarks on the importance of the work.

*1. I miss consideration of many of the factors omitted from the emission estimation procedure. This study basically used temperature, and agricultural statistics, to calculate emission factors (EFs). However, with respect to emissions from livestock/poultry, wind-speed is also a very important factor (e.g. Gyldenkaerne et al.,*

*2005, Skjoeth et al., 2011, Flechard et al, 2013). Many other factors should also impact NH₃ emissions, such as radiation, rainfall (and other precipitation), leaf-wetness, atmospheric stability, large uncertainties in the so-called Gamma factors, or bi-directional exchange in general (Bash et al, 2013, Flechard et al., 2013, Massad et al, 2010, Wichink Kruit et al., 2012). Consideration of such factors might also help to explain some if the model discrepancies outlined in Section 3, and should at least be considered before trying to explain all such discrepancies in terms of temperature and a few selected variables only*

**Response and revisions:**

We thank and agree the reviewer's important comment. In this work, we mainly compared the magnitude and the spatial and temporal distribution of the YRD NH₃ emissions estimated with two different methodologies, and evaluated the two inventories through air quality modeling based on available satellite and ground observation within the region. Compared to E1, in particular, E2 included the impacts of the growing and farming cycles, soil properties (pH) and selected meteorological factor (temperature) on NH₃ emissions for fertilizer using sector, and those of manure management processes and ambient temperature for livestock/poultry breeding. Besides the parameters we are concerned with, however, some other factors and processes also play important roles on atmosphere-land exchange of NH₃, as pointed by the reviewer. Those factors/processes that were not considered in this work include given meteorological factors (e.g., wind speed, precipitation and leaf surface wetness), surface layer turbulence, air and surface heterogeneous-phase chemistry, and plant physiological conditions (Flechard et al, 2013). In general, those factors/processes could be integrated in the bi-directional surface–atmosphere exchange module coupled in the air quality modeling, and improved estimation of NH₃ flux (emissions and depositions) were expected. The modeling system with the bi-directional NH₃ exchange were reported to be able to reduce the biases and error in simulation of NHx (NH₃ +NH₄⁺) wet deposition and ambient aerosol concentrations for both US and Europe (Bash et al, 2013; Wichink Kruit et al., 2012). Limited studies on the bi-directional NH$_3$ exchange were found for China (e.g., Fu et al., 2015). Out of the scope of current work, we did not focus on the bi-directional NH$_3$ exchange module and did not include the module for emission evaluation and comparison. We agree with the reviewer that the ignorance of given parameters/process in the estimation could potentially further explain the discrepancy between the simulation and observation. A more comprehensive evaluation and comparison in NH$_3$ emission inventories was thus suggested in the future, including the bi-directional NH$_3$ exchange and the top-down constraint with inversed modeling.

We have discussed this limitation and added relevant literatures **in lines 561-580, page 18 in the revised manuscript**.

*2.* *The authors use meteorology from ECMWF for their emissions, but why not the WRF model, since that is obviously available and is used for their CMAQ runs?*

**Response and revisions:**

We thank the reviewer's comment. We do not have very specific reason for using the ECMWF instead of WRF. When calculating the emissions, the underlying data open to the public were preferentially selected. ECMWF provided daily average data that satisfied our need of emission estimation and they were open to the public, thus we selected the dataset.

*3.* *The equations used are generally clearly written out, although it isn't always clear where they are coming from. For example, is it correct that equations 2 & 3 are a mixture of methods from Huang et al 2012 and EEA 2013? On the other hand, I read in various sections of EEA 2013 that temperature functions could not be provided (e.g. chap. 3.D crop production and agricultural soils) If from EEA, then it would also be good to cite the scientific papers underlying the EEA guidelines, and to be more specific as to which sections of EEA are being cited (it is a monster document).*

**Response and revisions:**

We appreciate the reviewer's comment. The specific EEA guidelines (EEA 2013a; 2013b; 2009) were provided in the revised manuscript. For Eq. 2 & 3, in particular, the linear relationships between $NH_3$ volatilization rate and temperature/soil pH were described in Chap. 4.D crop production and agricultural soils of EEA (2009)/Huang et al. (2012), and we specified them respectively **in lines 227-228, page 8 and lines 211-212, page 7 in the revised manuscript**.

*4. Some other points:*

*P2. The abstract is rather long, and should be shortened for clarity.*

**Response and revisions:**

We thank the reviewer's comment and the abstract was shortened.

*P3, L67. $NH_3$ is said to react with NOx, but NOx usually means $NO+NO_2$. I think the authors mean $HNO_3$?*

**Response and revisions:**

We thank the reviewer's reminder and it is corrected as nitric acid ($HNO_3$) in the revised manuscript.

*P3, L78-81. The sentence is a little unclear. Clarify.*

**Response and revisions:**

We thank the reviewer's comment. We mean that $SO_2$ and $NO_X$ emissions have gradually decreased due to improved control, thus the $NH_3$ emissions was found to play a greater role on the secondary aerosol formation and nitrogen deposition, compared to previous years. The sentence is rewritten **in lines 72-76, page 3 in the**

**revised manuscript**:

Recently the $SO_2$ and $NO_X$ emissions have gradually decreased due to implementation of various pollution control measures in China, thus $NH_3$ emissions were found to play a greater role on secondary aerosol formation and nitrogen deposition compared to previous years.

*P4, L112. Methods of including meteorology in $NH_3$ emissions have been around for some time and should be mentioned, e.g. Gyldenkaerne et al., 2005, Skjoeth et al., 2011, Wichink Kruit et al., 2012, Bash et al., 2013.*

**Response and revisions:**

We thank and agree the reviewer's comment. We have added the relevant papers and description **in lines 110-111, page 4 in the revised manuscript**.

*P5, L148. Another source of human-related $NH_3$ emissions is pets. As shown in e.g. Sutton et al 1995, 2000, human pets can be as significant as human metabolism with regard to NH3 emissions.*

**Response and revisions:**

We thank and agree the reviewer's comment. Due to lack of detailed statistic, we did not include pet emissions in current $NH_3$ inventories. Given the relatively small fraction in total emissions (e.g., less than 2% for United Kingdom by Sutton et al.), we believe that the uncertainty was limited. We have added the explanation **in lines 149-152, page 5 in the revised manuscript.**

*P6, L168. Using should be used.*

**Response and revisions:**

We thank the reviewer's reminder and it is corrected in the revised manuscript.

*P7, L187. Give reference for radiometer*

**Response and revisions:**

We thank the reviewer's reminder and the reference for radiometer is given in the revised manuscript (Davies et al., 2009).

*P7, L202. The study of Huang et al 2012 uses a linear relationship between pH and EF. Why is the relation here said to be near-linear?*

**Response and revisions:**

We thank the reviewer's reminder and it is corrected as linear in the revised manuscript.

*P7. What is the time-resolution of the EF calculations?*

**Response and revisions:**

The time-resolution of EF calculation is monthly. In the method, the fertilization method (top or basal dressing) was month-dependent, and monthly average temperature was applied for the EF calculation. We have added the information **in lines 212-213, page 7 and line 220, page 8 in the revised manuscript.**

*P8, L232. Surely fertilizer application at 15-20cm affects the pH of the soil; doesn't this affect the assumptions made when using global pH data from IIASA?*

**Response and revisions:**

We thank and agree the reviewer's comment. Previous domestic experimental studies in China (e.g, Zhong et al., 2006) indicated that the fertilizer application would increase the soil pH, particularly for the acidic soils. Bias thus existed in soil pH from the global database, without considering the detailed schedule and method of fertilizer application. As the quantitative relation between the fertilizer application and soil pH was still lacking at the regional scale in China, we ignored the interaction between the fertilizer application and soil pH in Eqs.(2). We acknowledged the limitation and added the explanation **in lines 243-248, page 8 in the revised manuscript.**

*P9. The basic references of the CMAQ model should be given, not just a web-address.*

**Response and revisions:**

We thank the reviewer's reminder and the basic operational guidance of CMAQ by University of North Carolina was provided in the revised manuscript (UNC, 2010).

*P10. Which version of MEGAN was used? Did you use data provided by Sindelarova, or did you use the MEGAN model itself? If the latter, a Guenther et al ref would seem more*

**Response and revisions:**

We thank the reviewer's comment. We used the MEGAN 2.1. The literature (Guenther et al., 2012) has been added in the revised manuscript.

*P10. Again, give reference to the model developers - this time for WRF.*

**Response and revisions:**

*We thank the reviewer's reminder* and the reference of WRF is provided in the revised manuscript (Skamarock et al., 2008).

*P11. The Lanciki 2018 reference for MARGA is missing.*

**Response and revisions:**

We thank the reviewer's reminder and the information of Lanciki (2018) is provided in the revised manuscript.

*P15. The citation of Wei et al (2015) is in Chinese, and thus not helpful for most authors. This instrument has been around for many years, and the artifacts documented elsewhere. Please find some citations in English for the problems mentioned.*

**Response and revisions:**

We thank the reviewer's comment and provided English papers for the problem **in lines 484-485, page 16 in the revised manuscript.** (Chen et al., 2017; Schaap et al., 2011; Stieger et al., 2018).

*P28, Use molecule not "mole.", to avoid confusion with the mole unit.*

**Response and revisions:**

We thank the reviewer's reminder and molecule is used in the revised manuscript.

*P31. Table 3. Correlation coefficients should be added, and the time-resolution of the statistics mentioned.*

**Response and revisions:**

We thank the reviewer's comment. The correlation coefficients between the observation and simulation were added in the revised Table 3, and the time-resolution of the statics was hourly, as mentioned in the revised caption of the table.

*P31 cont. for all Tables make it clear if statistics are calculated from hourly, daily or monthly values.*

**Response and revisions:**

We thank the reviewer's reminder. The statistics in Tables 3 and 6 were calculated based on the hourly values, and those in Tables 4 and 5 were from the daily values (the value of one hour (9:30am for satellite observation and the average of 9:00am-10:00am for simulation) per day). The information has been added **in the revised captions of Tables 3-6**.

*There are small English misses throughout, for example with regard to singular or plural, or omission of the definite article (the).*

**Response and revisions:**

We thank the reviewer's comment and the grammar errors are corrected in the revised manuscript.

**Response and revisions:**

We thank the reviewer's reminder and terms are corrected in the revised manuscript.

*L206-207 Are Tbasal and T0 in different units? Otherwise, 273.15 wouldn't be needed*

**Response and revisions:**

We thank the reviewer's reminder and 273.15 was deleted in the revised manuscript.

*L213 'method' -> 'application method'? (if I'm guessing correctly). What are the possible methods?*

**Response and revisions:**

We thank the reviewer's reminder and it is revised as application method (basal dressing).

*L347-L349 where is this shown about the EFs being from hot seasons?*

**Response and revisions:**

We appreciate the reviewer's comment. Most of the measurements on emission factor of ammonia from fertilizer application were conducted in summer or late spring (Cai et al., 2002; Huo et al., 2015; Su et al., 2006), especially those using micrometeorological method. It is expectable since that the basal dressing of single-season rice and maize as well as top dressing of wheat are usually conducted in late spring or summer. However, the crop rotation varies a lot in China, and part of the nitrogen-containing fertilizer actually is not applied in hot seasons. Emission estimation based on those emission factors may thus overestimate the emission intensity of ammonia (Huo et al., 2015; Wang et al., 2011; Zhang et al., 2010). We have provided relevant literatures and added the above discussion **in lines 361-369, page 12 in the revised manuscript.**

*L518-L519 Please reword. IASI is an instrument, so it cannot 'provide' an averaging kernel.*

**Response and revisions:**

We thank the reviewer's reminder and the sentence is rewritten **in lines 537-538, page 17 in the revised manuscript:**

As the ESPRI product of $NH_3$ VCDs we applied in the study does not provide the averaging kernel…

*Figure 1 caption: "Studying area and research domain" -> aren't study area and research domain the same?*

**Response and revisions:**

We thank the reviewer's reminder, and they are the same. We delete the "Studying area" in the figure caption.

*Figure 3 and Figure 6: 'Januray" -> "January"*

**Response and revisions:**

We thank the reviewer's reminder and the errors are corrected.

*Figure 4: emissions misspelled in the y-axis label Figure 4: Suggest giving fertilizer and livestock consistent colors, then keeping E1 as solid fill but E2 as hatched for easier reading*

**Response and revisions:**

We thank the reviewer's reminder and the figure is improved as required.

*Figure 6: colorscales could have greater consistency*

**Response and revisions:**

We thank the reviewer's reminder and the figure is improved as required.

*Figure 9: the subplots should have a consistent axis font size*

**Response and revisions:**

We thank the reviewer's reminder and the same font size is applied in all the subplots.

*Figure 10; from caption, shouldn't axis limits be same as Figure 9? Also, helpful to add the border lines like in Figure 9 so one is orientated where they are looking*

**Response and revisions:**

We thank the reviewer's reminder and revised the axis limits. The border lines have also been added in the revised Figure 10.

*Figure S4: there is one main cluster of data along the black line, but why is there seem to also be a second one? Also, what is the significance of the red dots, which do not fit well especially for the ABC panel?*

**Response and revisions:**

The black line is obtained through linear regression based on all the blue dots (including the "second cluster" mentioned by the reviewer).

Different from blue dots that are calculated for all the grids within the research domain of this study, the red dots are taken from available field measurement studies, as we mentioned in the figure caption. The gap between them, in particular at lower soil pH, explained the possible uncertainty in current method, i.e., the current linear assumption between the soil pH and $NH_3$ volatilization rate might not be appropriate for soil with low pH values for eastern China. We discuss it **in lines 553-560, page 18 in the revised manuscript.**

**3.1 Comparison between the two inventories**

Table 2 summarizes the $NH_3$ emissions estimated with E1 and E2 by source category and province for the YRD region in 2014. Agricultural activities (livestock farming and fertilizer) were identified as the most important sources of $NH_3$, with the fraction to total emissions ranged 74-84% in the two methods. Applying the constant emission factors, E1 derived a total $NH_3$ emission estimate 60% larger than that by E2 that characterized the agricultural processes. In particular, emissions from agricultural activities in E1 were calculated as twice of those in E2. At the national scale, similarly, Dong et al. (2016) applied the constant emission factors and estimated the total $NH_3$ emissions at 16.1 Tg for China, 64% larger than 9.8 Tg by Huang et al. (2012) with the agricultural processes characterized. The clearly larger estimation by constant emission factors was due partly to the fact that most domestic measurements on the emission factors of $NH_3$ from fertilizer application were conducted in hot seasons (late spring and summer), when the basal dressing of single-season rice and maize and top dressing of wheat were usually conducted (Cai et al., 2002; Huo et al., 2015; Su et al., 2006). However, the crop rotation varied a lot in China, and part of the nitrogen fertilizer was actually not applied in hot seasons. Emission estimation based on those emission factors may thus overestimate the $NH_3$ emission intensity (Huo et al., 2015; Wang et al., 2011; Zhang et al., 2010). 
[revised manuscript text omitted]

Limitation should be acknowledged in the emission comparison and evaluation.
Besides those we paid extra attention to in E2 (e.g., temperature, soil property,
fertilizer application method and manure management process), other factors could
also be influential on air-surface exchange of $NH_3$ and thereby $NH_3$ emissions,
including meteorology parameters (wind speed, precipitation, and leaf surface
wetness), surface layer turbulence, air and surface heterogeneous-phase chemistry,
and plant physiological conditions (Flechard et al, 2013; Gyldenkaerne et al., 2005,
Skjoeth et al., 2011). With those factors integrated in a bi-directional
surface–atmosphere exchange module in air quality modeling, the $NH_3$ emission
inventories were improved and the biases in simulation of $NH_3$ and $NH_4^+$ aerosol
concentrations were reduced for both US and Europe (Bash et al, 2013; Wichink Kruit
et al., 2012). The ignorance of given parameters/process in current work could thus
partly explain the discrepancy between the simulation and observation. Applying the
bi-directional $NH_3$ exchange module, for example, Wichink Kruit et al. (2012) found
increased $NH_3$ concentrations for agricultural source areas due to the elevated life
time and transport distance of $NH_3$ in the model. The result implied a possible
correction on the underestimation in $NH_3$ concentrations, as shown in Tables 3 and 4.
Therefore, a more comprehensive evaluation and comparison in $NH_3$ emissions was
thus suggested in the future, including the bi-directional $NH_3$ exchange and the
top-down constraint with inversed modeling.

[revised manuscript text omitted]
. The relations between environmental/meteorology conditions and $NH_3$ volatilization were not fully considered, and the bi-directional surface–atmosphere exchange was not included, resulting in bias in emission estimation. Uncertainties come also from the limitations in ground and satellite observation and incomplete mechanism of SIA formation in current air quality model. For better understanding the role of $NH_3$ emissions in regional air quality, more measurements on both sources and ambient concentrations are recommended in the future.

**Data availability**

The Multi-resolution Emission Inventory for China used in this study was obtained at http://www.meicmodel.org/ (last access: 31 July 2019, Tsinghua University, 2012). The high-resolution inventory for Jiangsu province was obtained in
* * *

Zhou et al. (2017) and can be accessed at http://www.airqualitynju.com/ (last access: 31 July 2019). The daily $NH_3$ VCDs measured through IASI was obtained from ESPRI data center at http://cds-espri.ipsl.upmc.fr/etherTypo/index.php?id=1700&L=1 (last access: 31 July 2019). The two $NH_3$ emission inventories developed in this work (E1 and E2) will be available with the publication of this paper at http://airquality.nju.com.

**Author contributions**

YZ developed the strategy and methodology of the work and wrote the draft. MY ran the model and produced the figures. XH revised the method and provided useful comments. FC and JZ conducted ground observation of $NH_3$ and aerosols.

**Competing interests**

The authors declare that they have no conflict of interest.

**Acknowledgements**

This work was sponsored by Natural Science Foundation of China (91644220 and 41575142) and the National Key Research and Development Program of China (2017YFC0210106). We would like to acknowledge Qizhen Liu and Zhong Zou from Shanghai Environmental Monitoring Center and Yunhua Chang from Nanjing University of Information Science & Technology for the ground measurement data, Qiang Zhang from Tsinghua University and Cheng Huang from Shanghai Research Academy of Environmental Science for emission data, and Simon Whitburn from Université Libre de Bruxelles and Yuanhong Zhao from Peking University for satellite data processing.

[revised manuscript text omitted]

**Table 5 The NMBs and NMEs between the simulated and observed daily NH₃ VCDs by soil pH and month**

| pH | Statistics(%) | January E_1 | January E_2 | April E_1 | April E_2 | July E_1 | July E_2 | October E_1 | October E_2 |
|---|---|---|---|---|---|---|---|---|---|
| pH>7.5 | NMB | 114.88 | 28.04 | 81.41 | -38.99 | 43.3 | 4.24 | 67.99 | 46.95 |
| | NME | 117.8 | 49.27 | 89.23 | 44.38 | 56.11 | 48.13 | 71.49 | 57.44 |
| 7.5<=pH<6.5 | NMB | 92.82 | 9.19 | 44.6 | -54.14 | 39.27 | -10.78 | 44.01 | 11.13 |
| | NME | 95.83 | 34.16 | 64.13 | 54.7 | 52.52 | 45.54 | 52.54 | 37.69 |
| pH<=6.5 | NMB | 41.61 | -11.76 | 1.30 | -67.41 | -12.43 | -55.81 | 8.64 | -25.48 |
| | NME | 54.72 | 36.76 | 60.16 | 68.5 | 34.78 | 56.72 | 35.27 | 43.68 |

**Table 6 The modeling performance at JSPAES and SHPD in cases with different SO$_2$ and NO$_X$ emission estimates. The NMBs and NMEs were based on the observed and simulated hourly concentrations.**

| | | JSPAES | | | SHPD | | |
|---|---|---|---|---|---|---|---|
| | Cases | Increased/ Decreased % | NMB % | NME % | Increased/ Decreased % | NMB % | NME % |
| NH$_3$ | Base case | | 1.73 | 56.94 | | 32.32 | 98.36 |
| | Case 1 | 10.14 | 11.09 | 59.02 | 11.67 | 47.54 | 102.68 |
| | Case 2 | -1.17 | -0.59 | 57.85 | -0.83 | 29.51 | 96.93 |
| | Case 3 | 8.48 | 9.29 | 59.64 | 11.12 | 44.92 | 100.94 |
| NH$_4^+$ | Base case | | 15.01 | 42.27 | | 84.45 | 100.23 |
| | Case 1 | -8.67 | 5.19 | 39.24 | -10.99 | 62.53 | 84.93 |
| | Case 2 | 1.87 | 17.55 | 45.40 | 1.40 | 87.40 | 102.37 |
| | Case 3 | -6.95 | 7.33 | 41.85 | -10.36 | 65.69 | 86.27 |
| SO$_4^{2-}$ | Base case | | 14.38 | 43.65 | | 91.92 | 110.18 |
| | Case 1 | -17.63 | -4.90 | 40.81 | -19.59 | 54.30 | 82.62 |
| | Case 2 | 2.76 | 18.42 | 43.7 | 1.55 | 94.34 | 112.30 |
| | Case 3 | -14.91 | -1.98 | 39.39 | -18.45 | 55.96 | 83.67 |
| NO$_3^-$ | Base case | | -6.55 | 44.81 | | 29.46 | 93.69 |
| | Case 1 | 1.25 | -5.92 | 44.52 | 6.30 | 37.56 | 92.51 |
| | Case 2 | 0.86 | -5.85 | 46.71 | -0.43 | 34.61 | 98.52 |
| | Case 3 | 1.85 | -4.90 | 46.51 | 5.78 | 42.85 | 97.19 |

**Figure 1**

[Figure]

**Figure 2**

[Figure]

**Figure 3**

[Figure]

(a) January                        (b) April

[Figure]

(c) July                        (d) October

**Figure 4**

[Figure]

**Figure 5**

[Figure]

(a) Fertilizer use in E1

(b) Fertilizer use in E2

(c) Livestock farming in E1

(d) Livestock farming in E2

(e) Total emissions in E1

(f) Total emissions in E2

**Figure 6**

[Figure]

(a) Fertilizer_January    (b) Total_January (c) Fertilizer_April    (d) Total_ April (e) Fertilizer_July    (f) Total_ July

[Figure]

(g) Fertilizer_October       (h) Total_ October

**Figure 7**

[Figure]

**Figure 8**

[Figure]

(a) SHPD_April (b) SHPD_July

[Figure]

(c) SHPD_October (d) JSPAES_October

**Figure 9**

[Figure]

(a) E1 January    (b) E1 April    (c) E1 July    (d) E1 October (e) E2 January    (f) E2 April    (g) E2 July    (h) E2 October

**Figure 10**

[Figure]

(a) E1 January   (b) E1 April   (c) E1 July   (d) E1 October (e) E2 January   (f) E2 April   (g) E2 July   (h) E2 October